



# The role of basal hydrology in the surging of the Laurentide Ice Sheet

William H. G. Roberts[1,2], Antony J. Payne[2], and Paul J. Valdes[1]

[1]BRIDGE, School of Geographical Sciences, University of Bristol
[2]Bristol Glaciology Centre, School of Geographical Sciences, University of Bristol

*Correspondence to:* William Roberts (william.roberts@bristol.ac.uk)

**Abstract.** We use the Glimmer ice sheet model to simulate periodic surges over the Laurentide Ice Sheet during the Last Glacial Maximum. In contrast to previous studies we use the depth of water at the base of the ice sheet as the switch for these surges. We find that the surges are supported within the model and are quite robust across a very wide range of parameter choices, in contrast to many previous studies where surges only occur for rather specific cases. The robustness of the surges is likely due

to the use of water as the switch mechanism for sliding The statistics of the binge-purge cycles resemble observed Heinrich Events. The events have a period of between $10 - 15$ thousand years and can produce fluxes of ice from the mouth of Hudson Strait of 0.05 Sv – a maximum flux of 0.06 Sv is possible. The events produce an ice volume of $2.50 \times 10^6$ km$^3$ with a range of $4.30 \times 10^6$ km$^3$ – $1.90 \times 10^6$ km$^3$ possible. We undertake a suite of sensitivity tests varying the sliding parameter, the water drainage scheme, the sliding versus water depth parametrization, and the resolution all of which support the ice sheet surges.

This suggests that internally triggered ice sheet surges were a robust feature of the Laurentide Ice Sheet and are a possible explanation for the observed Heinrich Events.

## 1   Introduction

Since the first discovery of distinct layers of ice raft debris (IRD) in North Atlantic sediment cores (Heinrich, 1988), debate has raged about the cause of these 'Heinrich Layers' and the 'Heinrich Events' (HE) that led to their deposition. These HE have

been implicated as the cause of global climate fluctuations (e.g. Broecker, 1994) because of the striking coincidence between the occurrence of Heinrich Layers and changes in a number of proxy climate records (see Hemming, 2004, and references therein). The ultimate cause of the HE is not clear however. Any mechanism to describe HE should be able to explain three key observations: the relatively short timescale over which the IRD layers form; the recurrence of the layers every 5–10 thousand years; the predominant source of the IRD as Hudson Bay. A number of mechanisms have been proposed that fit these criteria

Johnson and Lauritzen (1995) proposed that periodic discharges from an ice dammed lake in Hudson Bay, a jökulhlaup, could explain the presence of the Heinrich Layers. It has been proposed (Hulbe, 1997; Hulbe et al., 2004) that the collapse of an ice shelf in the Davis Strait would provide sufficient sediment rich icebergs to explain the layers. A number of authors (MacAyeal, 1993; Marshall and Clarke, 1997; Payne, 1995) have suggested that the Laurentide Ice Sheet (LIS) could produce large volumes of icebergs through an intrinsic instability in the ice sheet that gives rise to its periodic collapse: the "binge-purge" mechanism.



Finally a hybrid mechanism (Alvarez-Solas and Ramstein, 2011) again assumes that HE are the result of surges from the LIS, but suggests that the surges are paced by the collapse of an ice shelf in Davis Strait that buttresses the LIS. Of these only the binge-purge mechanism can explain both the volume of ice and the timing of the discharges implicitly, because both features are set by the geometry and composition of the ice sheet. This makes the binge-purge model an especially appealing model as it

does not require any external forcing to explain the timing of the events. Debate about the merits of both intrinsic binge-purge HE or externally-triggered HE continues. Neither explanation can be fully supported by the limited data that are currently available. Therefore until such time as these data do exist, it is not possible to say that one or other mechanism is correct. Indeed it is not even possible to say that there is one single cause for all Heinrich Events.

Uncertainties in data supporting the binge-purge mechanism include the unexplained coincidence of Heinrich Events with

some of the coldest, longest stadial periods (Bond and Lotti, 1995). If such a link does exist it is difficult to understand how features of the surface climate will express themselves in a phenomenon that takes place at the ice sheet bed. Furthermore, if HE are the result of a regular oscillation in the ice sheet over Hudson Bay, then we would expect that the sediments in all Heinrich Layers would have a signature of Hudson Bay: they do not (Hemming, 2004; Naafs et al., 2013).

Uncertainties surrounding an external trigger include the ultimate reason for the warming beneath the assumed ice shelf

covering parts of the Labrador Sea. Although changes in the Atlantic Meridional Overturning Circulation (AMOC) have been implicated as the cause for the warming (e.g. Marcott et al., 2011; Menviel et al., 2014), it is not clear why the AMOC is itself reduced. If we assume that AMOC reduction goes hand in hand with Dansgaard/Oeschger (D/O) events, which is itself by no means certain (Dokken et al., 2013), it must be explained why the AMOC is more reduced during some D/O events than others such that a HE does not occur for each D/O event. This could arise from the link between the coldest stadials and HE.

If temperatures are anomalously cold we would expect a reduction in the mass lost from the ice sheet from surface melt but an increase in the mass lost due to calving. With no net change in the amount of mass lost from the ice sheet this represents only a change in the mechanism by which mass is lost (Marshall and Koutnik, 2006). An increase in the calving could make it easier for the freshwater from the ice sheet to impact the AMOC, but it will undoubtedly also increase the ice shelf's thickness making it more resistant to melt and a better buttress. Such changes in the ice shelf thickness have not been simulated (Hulbe,

1997). Other key features required for an external trigger also remain, so far, unobserved. Not all HEs are observed to have an associated subsurface warming, although this is due to a lack of observations rather than an evidence of absence (Marcott et al., 2011). There is also no evidence for an ice shelf in the Labrador Sea. The geography of the Labrador Sea makes it likely that an ice shelf would form there, however its size and therefore capacity to buttress the ice sheet is unknown. Observations of this ice shelf are key to supporting this mechanism.

One often overlooked feature that also needs to be explained by either mechanism is the evidence of IRD layers from other parts of the LIS (Stokes et al., 2005). If the ice sheet collapses due to internal dynamics this result is relatively easy to explain; if the ice sheet must be forced we need to find another external trigger mechanism. In particular, such external forcing would require ice shelves to exist in other sectors of the ice sheet, areas which may be less conducive to their formation that the Labrador Sea.





Given these long lists of problems with both, and we stress both, mechanisms for the cause of HE, it is not possible to claim pre-eminence, on the basis of data, for one mechanism over another. In this manuscript we shall address the binge-purge mechanism and show that oscillations are still possible when basal hydrology is better simulated, and that the surges are not the result of numerical instabilities brought on by a particular treatment of ice sheet motions at the bed.

At its most basic, the binge-purge mechanism draws upon the idea that over Hudson Bay and Hudson Strait the LIS moves in one of two modes: slow deformational creep when the ice sheet is frozen to the bed, and rapid sliding when the ice sheet bed is melted and well lubricated. The ice sheet flips between these two states as a consequence of internal sources and sinks of heat. This mechanism has been found to operate in both box models (MacAyeal, 1993) and a number of simple 2-D ice sheet models (Payne, 1995; Fowler and Schiavi, 1998).

MacAyeal (1993) modelled the transition between the sliding and creeping state as a function of the temperature at the bed of the ice sheet: if the base of the ice sheet was at pressure melting the ice sheet slid, below pressure melting it crept. In this model the base of the ice sheet was warmed by the geothermal heat flux until the bed was at pressure melting at which point it began to slide. Heat generation from friction within the ice sheet then kept the temperature high until cooling from cold ice in the ice sheet finally brought the base below pressure melting at which point the sliding stopped. Payne (1995) found a similar

result but in his model the warming at the base prior to sliding was due to strain heating within the ice sheet. The importance of strain heating in prompting an ice sheet to slide in HE was also proposed by Verbitsky and Saltzman (1995). Fowler and Schiavi (1998) also found surging behavior in their 2-D model, however in their case the switch between sliding and creeping was based on the depth of water at the base of the ice sheet. Strain heating was also of great importance in the development of the surges and they noted the backward progression of a wave of strain heating in the ice sheet and coined the term the

'activation wave' for this swift initiation of ice sheet sliding. A common feature of all of these models is a sawtooth pattern in the height of the ice sheet with a slow binge phase, followed by a much faster purge phase.

Although the presence of surges with very similar properties to real HEs in simple models supports the validity of the binge-purge mechanism, given the large number of parameterizations in these models it is not impossible that these surges are the result of the approximations. Only using more complex models can this mechanism be verified. For this reason a number of

more complete 3-D ice sheet models have also been used to simulate HE from the LIS.

Marshall and Clarke (1997) found surging behavior in their 3-D model of the LIS and in an extension of this, Papa et al. (2005) found that the surges were initiated by the strain heating at the base of the ice sheet, although we note that Papa et al. (2005) did not find surging behavior in Hudson Bay and Hudson Strait in contrast to the original Marshall and Clarke (1997) study. Calov et al. (2002) found that a 3-D shallow ice approximation (SIA) model of the LIS with a switch based on the basal

temperature would surge with a period close to that observed. The exact mechanism causing the surges in this model was not made clear. More recently Calov et al. (2010) used an idealized representation of the Hudson Bay and Hudson Strait to test the ability of a number of current 3-D ice models to simulate Heinrich-like events. Only a small subset of the models produced binge-purge cycles, and in this subset only a yet smaller subset produced surges that drained the ice from the centre of Hudson Bay as in real HE. The switch from creeping to sliding in these models is based upon the temperature at the base of the ice

sheet. Therefore although the binge-purge mechanism is quite robust in 2-D models, its presence in more complex 3-D models



is not assured. Furthermore the only switch from creeping to sliding that has been used is based on the basal temperature of ice sheet.

One major issue with the 3-D modelling studies of HE is the use of the shallow ice approximation. As discussed by Hindmarsh (2009), ice streams can only be accurately simulated by an ice sheet model that includes longitudinal stresses. Ice

streaming, which is associated with the rapid loss of ice in HE, is model resolution dependent in shallow ice models and, in most cases, the streaming is not obviously the result of some physical process rather it may be the result of a numerical artefact. The hybrid shallow ice/shallow shelf model of Álvarez Solas et al. (2011) tried to alleviate this problem by incorporating the effect of horizontal stresses. However, intrinsic oscillations were not found to be present in this model and it required an external forcing to trigger any surging events.

Another issue with many of the previous 3D modelling studies has been the temperature switch mechanism that has been used. The basal traction in an ice sheet is strongly affected by water and its drainage. Drainage systems are generally considered as either efficient, low pressure, channelised systems (Röthlisberger, 1972; Nye, 1973), or high pressure water films (Weertman, 1966). Although water films are generally unstable, ultimately collapsing to a channelised system, in the presence of a sufficiently rough bed they can remain stable (Creyts and Schoof, 2009), furthermore, there is evidence that high water pressure

systems exist beneath ice sheet (Engelhardt and Kamb, 1997). The depth of water beneath ice sheets has been argued to be intimately related to the speed with the overlying ice can slide (Budd and Jenssen, 1987; Le Brocq et al., 2009). Therefore when considering the onset of sliding during HE, water must play a very important role. Most studies examining HE have neglected this, choosing instead to switch the model of ice sheet motion on the basis of temperature alone. Although water and basal temperature are intimately related – without the base of the ice sheet being at pressure melting no water will be

present – the assumption that the ice sheet will start to slide as soon as pressure melting is reached, regardless of the amount of water at the bed is not good. Thus a better way of considering the onset of sliding is to consider not only whether water is present at the ice sheet bed but also how deep this water is. This has been considered in both 2D (Fowler and Schiavi, 1998) and 3D (Kyrke-Smith et al., 2014) ice sheet models but in both cases the model set up was rather idealized. Kyrke-Smith et al. (2014) showed that with their 3D ice sheet model coupled to a basal hydrology model fast flowing ice streams could develop.

However, in their simulations once the ice streams were established, there was no mechanism to stop them, due to the ice sheet bed being kept at pressure melting. Thus, although 3D ice sheet models have been coupled to basal hydrology models, their configuration has never been realistic enough to show the importance of water in ice sheet surges.

In this study we shall simulate HEs in a 3-D ice sheet model, Glimmer (Rutt et al., 2009), using such a water scheme. We investigate what mechanisms give rise to the events and investigate how robust they are to changing a number of parameteri-

zations within the model. Unlike other studies using 3-D ice sheet models we shall use a switch mechanism similar to that of Fowler and Schiavi (1998), based on the water depth at the base of the ice sheet. We show that our model can simulate realistic HE but that the events are rather too long lived. In a series of sensitivity tests we show that although the exact behavior of the events does depend upon the parameterizations in the model, the presence of the events is not sensitive to these parameterizations nor the resolution of the model: the only necessary condition is the presence of water at the base of the ice sheet. We



address whether the surges are the result of numerical instability from using the SIA by running the model at progressively finer resolution to see if the nature of the events are resolution dependent: they are not.

In Section 2 we describe some features of the ice sheet model that we shall use; in Section 3 we describe how the model simulates the HE. Section 4 describes how sensitive the events are to a number of parameterizations and structural features

in the model. These are the sliding parameter (Section 4.1), the water drainage scheme (Section 4.2), the sliding versus water depth relationship (Section 4.3) and the model horizontal grid resolution (Section 4.4). In Section 5 we discuss how the HE simulated in this study agree with observations and other modelling studies. Finally in Section 6 we conclude.

## 2    Model Description

We use the Glimmer ice sheet model (Rutt et al., 2009) which is a thermomechanical ice-sheet model that uses the shallow

ice approximation (SIA) (e.g. Hutter, 1983; Hindmarsh and Le Meur, 2001). For more details on this model we refer the reader to Rutt et al. (2009), and concern ourselves below with some features of the model that are neccessary for subsequent discussions.

The SIA neglects longitudinal stress gradients. Although these stresses are negligible in the interior of a slow moving ice sheet, they are important at the margins where they are integral to ice shelf and ice stream dynamics. Furthermore, in regions

where horizontal shearing is important, for example at the boundary between slow moving ice and fast moving ice streams, longitudinal stresses are not negligible (Hindmarsh, 2009). The lack of longitudinal stresses in regions of high horizontal shear is of concern since we would expect such areas of high shear to occur during surging events when parts of the ice sheet are moving at relatively high velocities whilst surrounded by areas of much more slowly moving ice. We acknowledge this omission but must neglect it since using higher order approximations make the long model integrations that we need to perform

computationally impossible.

The time evolution of temperature in the ice sheet is determined by a balance of heating terms that represent: vertical diffusion, horizontal advection, internal heat generation or strain heating and vertical advection. At the base of the ice sheet the vertical gradient of temperature, contained in the vertical advection and diffusion terms is a result of heating by the geothermal heat flux and heating due to friction at the bed. A fuller mathematical description of these terms can be found in Rutt et al.

25  (2009)

### 2.1    Sliding Law

As described in the introduction surging events are the result of the ice sheet flipping between two states: creeping whilst frozen to the bed and sliding whilst on a lubricated bed. To switch between these two states we require a parameterization. Previous models have taken as the switch the temperature at the bed of the ice sheet (Calov et al., 2002, 2010; Papa et al., 2005). When

this temperature reaches the pressure melting point, it is assumed that the base of the ice sheet at that grid point has melted and therefore the ice sheet can slide on the lubricated bed. This switch assumes that the whole of a grid box is either frozen to the bed or is sliding. With a 50 km grid, each grid box represents 2500 km$^2$, therefore using such a switch assumes that the whole



2500 km$^2$ instantaneously transitions from stuck to sliding: this is obviously unrealistic. We therefore use a different switch that allows each grid box to progressively start sliding. This is based on the depth of water at the base of the ice sheet.

Following Le Brocq et al. (2009) we model the onset of sliding as a *tanh* function of water depth which has been shown to reasonably simulate sliding at the base of the present day West Antarctic Ice Sheet. This function takes the form of

$$C = C_o \left[ 0.5 + 0.5 \cdot tanh \left( \frac{(d-b)}{a} \pi \right) \right] \tag{1}$$

where $C_o$ is a sliding parameter and $d$ is the water depth. $a$ sets the depth of water over which the transition from sliding to stuck occurs and $b$ sets the depth at which the ice sheet sliding parameter $C = \frac{C_o}{2}$. By varying these parameters we can change the water depth at which sliding occurs and how quickly this transition happens. In this study the default values of $a$ is 0.8 mm and $b$ is 2.0 mm. As we have no *a priori* knowledge of what values these parameters should take we undertake an extensive sensitivity test to examine these parameters in Section 4.3. We find that the surging behaviour is robust over a large range of

parameters within the range of water depths that the model simulates.

Once the ice sheet has begun to slide the basal sliding speed, $\mathbf{u}$, in the ice sheet model is linearly related to the horizontal shear stress $\tau$ according to the sliding law:

$$\mathbf{u} = C\tau. \tag{2}$$

The speed scales according $C$, defined in Eq. 1, which depends upon a sliding parameter $C_o$. This takes one of two values, a very low value, 0.005 m Pa$^{-1}$ yr$^{-1}$, over regions of hard bed, and a higher value, 0.1 m Pa$^{-1}$ yr$^{-1}$, over regions of deep

sediment (for a typical 20kPa driving stress these give speeds of 100 m yr$^{-1}$ and 2km yr$^{-1}$). This assumes that in regions, such as the Hudson Bay, where there are deep layers of sediment, it is possible for the ice sheet to slide at much higher velocities than over a hard bed. This is because the ice sheet bed is more effectively lubricated (Clarke, 1987). These two regions are determined using a global sediment thickness map (Laske and Masters, 1997). In the Hudson Strait region we adjust the sliding parameter map to remove a slight kink in the Strait. This can be most clearly seen in the maps of basal velocity (Figs. 7 and

11). This has no effect on the presence of surges (again compare Figs. 7 and 11) but makes the analysis of the flow line far easier. We take 0.1 m Pa$^{-1}$ yr$^{-1}$ as the default value (this lies in the middle of the range of this parameter from previous studies such as Marshall and Clarke (1997); Calov et al. (2002); Papa et al. (2005); Calov et al. (2010)) however we also undertake a sensitivity test to examine how import this parameter is in setting the nature of surges in Section 4.1. We find that the surging behaviour is not at all sensitive to the exact value of this parameter.

## 2.2 Subglacial drainage

In this study we use the water depth as the switch for the sliding of the ice sheet therefore we need to simulate the drainage of water beneath the ice. We model this by assuming that no water drains into the bed beneath the ice sheet rather it is forced between the bed and the ice sheet, as if between two parallel plates, by differences in water pressure (Weertman, 1966; Le Brocq et al., 2009). We shall refer to this as the Water Sheet Scheme. Although this type of water scheme is generally unstable

Creyts and Schoof (2009) have shown that such a drainage system can be stable so long as there are sufficient protrusions to





support the base of the ice sheet, especially if the ice sheet is moving fast and water pressures are relatively low. Complete details of the formulation and implementation of this scheme are detailed in Le Brocq et al. (2009), we present here only the salient details necessary for the following discussion.

In this scheme we assume that the time rate of change of water depth, $d$, is given by

$$\frac{\partial d}{\partial t} = q - \nabla \cdot \mathbf{u_w} d \tag{3}$$

where $q$ is the melt rate $\mathbf{u_w}$ is the depth averaged water velocity calculated assuming laminar flow between two plates driven by differences in pressure. This is calculated by,

$$\mathbf{u_w} = \frac{d^2}{12\mu} \nabla \Phi. \tag{4}$$

The gradient of geopotential, $\nabla \Phi$ is calculated assuming that the water pressure is equal to the overburden pressure:

$$\nabla \Phi = \rho_i g \nabla S + (\rho_w + \rho_i) g \nabla h. \tag{5}$$

$S$ the surface elevation of the icesheet, $h$ the bed elevation, $\mu$, $\rho_{i,w}$ and $g$ are all constants. We assume here that the effective pressure is zero (see, e.g., Budd and Jenssen, 1987; Alley, 1996). Although we would expect the effective pressure to have an

impact upon the rate of sliding we neglect this effect as it is small.

Following Le Brocq et al. (2009) we assume that in Eq. 3 the water depth is in a steady state, $\frac{\partial d}{\partial t} = 0$, since the rate of flow of water is many orders of magnitude faster than the rate of flow of ice. We take a flux balance approach to solve,

$$\psi_{out} = \psi_{in} + q r^2 \tag{6}$$

where $\psi_{out}$ and $\psi_{in}$ are the fluxes out of and into the box, respectively, and $r$ is the grid box length. This finally leads to the equation

$$\mathbf{u_w} d = \frac{\psi_{in} + q r^2}{l} \tag{7}$$

where $l$ is the unit width of the grid cell, which depends upon the direction of flux through the gird box. $\psi_{in}$ is calculated using the flux routing scheme of Budd and Warner (1996).

From Eq. 7 we see that the flux of water out of the grid box (left hand side), is determined by both the melt rate (second term on right hand side) and the flux of water entering the grid box from upstream (first term on the right hand side). The depth of the water sheet that arises from this flux is then determined by the local gradient of geopotential, Eq. 4.

This parametrization has been proposed as a good approximation for use in modelling water under the West Antarctic Ice Sheet (Le Brocq et al., 2009). It has been suggested that its use could reproduce the ice surface morphology, the velocity and thermal regime within the ice sheet more accurately than current models.

Another, simpler, description for rate of change of water depth, $d$, under the ice sheet assumes that it is determined entirely locally as:

$$\frac{\partial d}{\partial t} = q - \frac{d}{\lambda} \tag{8}$$





where the basal melt rate, $q$, is a function of the temperature at the base of the ice sheet, and $\lambda$ is a specified timescale for water to drain through the bed. We refer to this as the Local Water Scheme. In essence this assumes that water drains either in large channels beneath the ice sheet or directly into the bed, thus water generated in one region has no influence on adjoining regions. We propose this scheme less as an accurate model of sub glacial drainage, more as a test of our model set up, in order to investigate how sensitive the model is to the exact details of the sub glacial drainage scheme.

In reality the routing of water beneath an ice sheet is far more complex than either of these two parameterizations. They do however represent the two ends of the continuum of ways that water might drain from under an ice sheet. As we shall show the surging behaviour occurs regardless of the water scheme we use, therefore we would argue that what is crucial for the surging is the presence of water not the exact details of how it is drained. We do however feel that the Water Sheet Scheme is a more representative scheme than the Local Water Scheme.

## 2.3 Further model details

In this study we run the model on a grid with a resolution of 50km in the horizontal and 11 levels in the vertical. Glimmer uses a sigma coordinate system in the vertical so the thickness of each level varies depending upon the total ice sheet depth. There is a concentration of levels at the base of the ice sheet where high resolution is more important.

Although Glimmer does allow for the use of a lithosphere model beneath the ice sheet, in order that we can make direct comparisons between the different runs in the suite of sensitivity tests, we uses a topography beneath the ice sheet that does not vary in time. For this we use the ice5g topography for 21ka (Peltier, 2004)

All model runs start with the ice5g ice distribution (Peltier, 2004) which is then allowed to freely evolve using the accumulation and surface temperature from the climate model and the specified basal sliding distribution. The models are allowed to reach a dynamic equilibrium over 50,000 years, and are then run for a further 100,000 years, the period over which the analysis is made. Diagnostic fields are output every 100 years. However when diagnosing the processes responsible for the surges we use output derived every time step.

To force the model the same constant climate forcing is used in all of the runs. This forcing is taken from an LGM run of the FAMOUS climate model. The surface mass balance used by the ice sheet model is calculated using the precipitation and temperature fields from the climate model and use a simple positive degree day scheme (Reeh, 1991; Rutt et al., 2009). The resulting surface mass balance field is shown in Fig. 1(a). The temperature field that forces the upper surface of the ice sheet is also shown, in Fig. 1(b). The base of the ice sheet is forced with a spatially and temporally constant geothermal heat flux that takes a value of $4.2 \times 10^{-2} Wm^{-2}$.

## 3 Simulated "Heinrich Event"

In this section we describe the structure and behavior of the HE in the default configuration of Glimmer. This configuration uses the Water Sheet Scheme, a sediment sliding parameter of 0.1 m Pa$^{-1}$ yr$^{-1}$, and a hard bed sliding parameter of 0.005 m Pa$^{-1}$ yr$^{-1}$. Firstly we describe the mean state of the ice sheet.



## 3.1 Equilibrium Ice Sheet

We recall that Glimmer is initialized from the ice5g ice sheet (Peltier, 2004), but it is then allowed to freely evolve in response to the bed topography and climate forcing. Following a spin up of the model for 50,000 years the model has reached a dynamic equilibrium where the ice sheet area and volume oscillate about a constant mean state: these oscillations are the modelled HE. The equilibrium mean area of the ice sheet, shown in Fig. 2, is $1.67 \times 10^7$ km$^2$ with a standard deviation of $2.21 \times 10^5$ km$^2$. This compares well with the ice5g distribution that the model was initialized with, which has an area of $1.68 \times 10^7$ km$^2$. The agreement is very good with only a slight southward extension of the Glimmer ice sheet in comparison to ice5g (compare the red and green lines in Fig. 2(a)) .

The mean volume of the Glimmer ice sheet is $4.16 \times 10^7$ km$^3$ with a relatively large $1.57 \times 10^6$ km$^3$ standard deviation, the result of the surging events. This is somewhat higher than the ice volume estimate of ice5g at $3.24 \times 10^7$ km$^3$. We see in Fig. 2 the spatial distribution of this ice. Spatially there are two distinct domes to the ice sheet, in agreement with the LGM ice sheet geometry of  Dyke and Prest (1987), both with heights of greater than 3500 m. Therefore, the larger volume of ice in Glimmer compared to ice5g is the result not of an ice sheet that has a taller peak than ice5g, but of one that is flatter, but on average thicker over much of its area.

Figure 2 also shows, in the filled contours, an expression of the variability in the ice sheet. This is the difference between the ice sheet thickness for composites of before and after all the HE in the 100,000 year model run (these are the events shown in Fig. 3). This shows that the ice sheet over the whole area of Hudson Bay is associated with HE. Hudson Bay is not the only source of variability within the ice sheet, however. There is also much variability in the ice sheet near both the southern and the north western margins. Significant variability in these areas is not unexpected as there is observational evidence for variability in the extent and volume of the ice sheet here (Stokes et al., 2005; Mooers and Lehr, 1997). However, in this paper we shall confine our discussion to the variability of the ice sheet over Hudson Bay and out of Hudson Strait.

We have shown that our configuration of Glimmer can realistically simulate the mean Laurentide Ice Sheet at the LGM. We have also shown that there is variability over Hudson Bay. We will now look in more detail at this variability and how it evolves over time.

## 3.2 Temporal Characteristics

Figure 3 shows the average height of the ice sheet over the centre of Hudson Bay and the flux of ice out of Hudson Strait. In the upper curves we see that the average ice sheet height over Hudson Bay exhibits the "saw tooth" that has been associated with thermomechanical surging events (Payne, 1995; MacAyeal, 1993). It shows a slow build up of ice over around 10,000 years, followed by a rapid reduction in ice sheet thickness over about 2000 years. Concurrent with the decrease in ice sheet height, is a large increase in ice flux or iceberg calving leaving the mouth of Hudson Strait. This flux lasts for around 3,000 years and has a peak flow of around 0.05 Sv. Looking at the high temporal resolution event, shown in the right panel, we can resolve more detail in the events.



We see that the flux of ice from the mouth of Hudson Strait, leads the decrease in Hudson Bay ice sheet height. Ice fluxes, of $0.02 - 0.03$ Sv, begin to flow from the mouth of the Hudson Strait about 1000 years before the ice sheet over the centre of the Hudson Bay registers a change in height. Indeed over this time the height of the ice sheet over Hudson Bay is increasing. This is the time it takes for the activation wave (Fowler and Schiavi, 1998) to propagate the length of the Hudson Strait into the

interior of the ice sheet. Once this activation wave reaches the centre of the Hudson Bay, the flux of ice out of Hudson Strait further increases to its peak of around 0.05 Sv, as this much larger source of ice is tapped. As the event develops the volume of ice leaving Hudson Strait decreases from its initial peak until, about 3000 years after the beginning of the event, the flux of ice abruptly stops and the event ends. This cycle repeats itself approximately every 13,000 years.

During each event around $2.50 \times 10^6$ km$^3$ of ice will have left the Hudson Strait and the central Hudson Bay ice sheet

thickness will have decreased by around 950 m.

All of these gross statistics, which are summarized in Table 1 along with estimates of similar statistics from previous studies, show that the variability of the ice sheet in Hudson Bay in Glimmer bears all of the hallmarks of HE discussed in the introduction. We shall defer a full discussion of how our model compares to these other estimates until Section 5 and turn our attention to the mechanics of how the events develop.

### 15 3.3 Anatomy of the Heinrich Events

Before the surge begins much of the interior of the ice sheet is at pressure melting point (Fig. 4). This means that even before the event begins, there is water at the base of the ice sheet (Fig. 6). However, because the depth of the water is low the ice sheet can not slide at its full speed. Furthermore, in Hudson Strait there is a region in which the ice sheet bed is still frozen, which not only prevents the ice sheet from sliding but also prevents the water that lies beneath the ice sheet in Hudson Bay from flowing

out.

This state continues until this region that is frozen to the bed can be melted. At the margin of the ice sheet there are a succession of small accelerations of the ice sheet. These are caused by the build up of water locally under the ice sheet which allows the ice to start sliding in response to the driving stress. However because the upstream ice is not sitting atop water it is unable to slide, thus the driving stress at the margin drops and the ice stops flowing. Each of these accelerations leads to an

increase in the gradient of the ice sheet surface farther inland which in turn leads to an increase in the strain heating inland. However, until the base of the ice sheet has warmed enough that this strain heating can melt the ice sheet base and produce a sufficiently deep water layer to allow fast sliding, the margins of the ice sheet will continue to pulse, but the interior of the ice sheet will remain stationary. We must note that although this frozen region in the mouth of Hudson Strait is common among all of the simulated HE, such a frozen region is not present in the surges that occur in other regions of the ice sheet. Thus, although

the frozen region is a feature of our HE it is not necessary for it to exist in order for surges to occur.

At some time the base of the ice sheet will warm sufficiently that the gradient in ice sheet surface, and its associated strain heating, can warm the interior of the ice sheet above pressure melting point. At this point a full blown surge can get under way. Time series of the terms that allow this to happen are shown in Fig. 5.



We see that before the ice sheet starts to slide, there is a small amount of water at its base, but it is not deep enough to allow the maximum amount of sliding. When the oceanward part of the ice sheet does begin to slide there is a peak in the gradient of the ice sheet surface which propagates backwards through the ice sheet into the interior. This can be seen in the contours on Fig. 6. This gives rise to an increase in the strain heating, which acts to increase the melt rate at the base of the ice sheet. This

increase in the melt does not, however, cause an immediate rise in the water depth at this location.

We recall that the water sheet depth is related to $\nabla\Phi$ through Eq. 4. Thus the increase in the surface gradient that gives the increased melt rate and brings on the surge also serves to route the water away from this region. This can be seen in Fig. 5 where there is a net flux of water away from the region (green line, middle panel). The routing of water away from the region is not large enough to prevent the build up of water at the base of the ice sheet, but it is large enough to delay it. As the water

depth increases the sliding speed increases and thus the heating rate from friction can increase. Friction becomes the dominant heating term as the strain heating begins to decrease (red line, top panel) and increases in a positive feedback with the water depth: more frictional heating increases the melt rate which increases the water depth which increases the sliding speed and so on. This feedback becomes stronger as the peak of $\nabla\Phi$ passes and the water is no longer routed away with the same intensity. Indeed after the activation wave has passed the water routing scheme acts to *increase* the water depth at the point in Fig. 5,

because it routes the water away from the inland regions, where the activation wave is, to the downstream regions. This can be seen in the pulse of water into the region (Fig. 5).

The progress of the activation wave and its effect on the basal water depth can be seen in the red contours in Fig. 6. In panel (c), at time 28,000, oceanward of the activation wave are deep water depths, inland the water is shallow enough that the ice sheet is not able to slide much. Once the activation wave has passed down Hudson Strait and across Hudson Bay, by about

28,200 in the figures, the whole of the basin is on enough water that it can start to fully slide. We see in Fig. 7 that when the ice sheet is in this mode the whole of Hudson Bay is sliding with speeds between 1 and 10 km yr$^{-1}$

As the event progresses the driving stress decreases, thus sliding velocities drop. As the velocity drops the frictional heating drops, and as this is the largest heating term the melt rate is also reduced. The event comes to a halt when the heating rate drops sufficiently that the melt rate can no longer produce enough water to maintain fast sliding. This is not the whole of the story

however as the water routing scheme also plays a part.

In general in the interior of the ice sheet there is a flux of water towards the margins, this leads to shallower water depths than the melt rate alone would predict. By contrast at the margins there is a general flux of water into these regions. Therefore in the interior of the ice sheet the water depth falls below the level needed to maintain the ice sheet sliding sooner than if the water were not routed away. When the interior does stop sliding, the water depth at the margin of the ice sheet falls quickly,

due to the lost water source, and it too begins to stop sliding. In this way the surge stops relatively quickly across the whole ice sheet together (in comparison to a surge in which the water is not routed from the interior to the margins).

In this section we have shown that surges can exist in our model. We have seen that they do not begin the moment that the ice sheet reaches pressure melting point, as the base of the ice sheet over much of Hudson Bay is at pressure melting for many years before the surge gets under way. Rather it is the presence of water, deeper than a certain depth at the base of the ice sheet

that initiates the surges. The details of the water scheme that we use give rise to a number of interesting features of the surges.



**Table 1.** Table of vital statistics of Heinrich Events from various studies

| Study | Period (yr) | Duration of Surge (yr) | Total Ice Volume ($10^4$ km$^3$) | Peak Iceberg Flux (Sv) |
|---|---|---|---|---|
| Hemming (2004) | $\sim 7000$ | $495 \pm 255$ | $3 - > 500$ | $0.25 - 1.0$ |
| Roche et al. (2004) | | | 85.8 | 0.29 |
| MacAyeal (1993) | 7260 | 450 | 125 | $0.16 - 0.08$ |
| Dowdeswell et al. (1995) | | 250-1250 | 27.0 | 0.02 |
| Marshall and Clarke (1997) | 5250 [b] | 750 [c] | 8.5 [d] | 0.004 [e] |
| Hulbe (1997) | | | 75 | |
| Hulbe et al. (2004) | | | $2.8 - 20$ | |
| Calov et al. (2002) | 4000-8000 | $<1000$ | $\sim 200$ | $0.1 - 0.2$ |
| Papa et al. (2005)[a] | 9000 | | | 0.04 |
| Roberts et al. (2014a) | | | $60 \pm 30/120$ | |
| Present Study | 11,000 | 2600 | 250 | 0.05 |

[a] No oscillations in Hudson Bay/Strait. [b] Ranges from $600 - 22,000$. [c] Ranges from $105 - 3260$. [d] Ranges from $1.6 - 24.2$. [e] Ranges from $7.5 \times 10^{-4} - 0.01$

The beginning of the surge is slightly retarded by the routing of the water away from the region of the activation wave. The end of the surge is slightly earlier than it might be and its demise faster because water is routed away from the interior of the ice sheet to the margins, thus co-ordinating the whole ice sheet. As in any modelling study we have had to make a number of assumptions about the physical parameterisations that we use. It could be argued that the surges that we see are the result of a peculiar set of these parameters. In the following section we investigate how sensitive our results are to these assumptions.

## 4 Parameter Sensitivity

### 4.1 Sliding Parameter

Previous studies, (Calov et al., 2002; Papa et al., 2005; Calov et al., 2010) found that the presence of surges was rather sensitive to the values of the sliding parameter over sediment rich areas. We recall that the sliding parameter, $C$, relates the speed at the base of the ice sheet to the driving stress (Equation 2). Calov et al. (2002) tested a range of sliding parameters and found that binge-purge events occurred for $C = 0.03 - 0.1$ m Pa$^{-1}$ yr$^{-1}$. At the low end of this range the events were rather small and irregular: larger values of $C$ produced larger more regular events. Papa et al. (2005) used a value of $C = 0.04$ m Pa$^{-1}$ yr$^{-1}$, and found that with values greater than this numerical instabilities developed in the model. The behavior of the model for $C < 0.04$ m Pa$^{-1}$ yr$^{-1}$ was not reported. Calov et al. (2010) investigated $C = 0.01, 0.02, 0.05, 0.1$ m Pa$^{-1}$ yr$^{-1}$ and found that in their idealized model set up, changing the sliding parameter had the effect of removing the surging behavior in some model runs. In those models where surging persisted at larger values of $C$, larger $C$ led to larger events with little change in their period.





To test the sensitivity of Glimmer we vary $C_o$, the soft bed sliding parameter, in Eq. 1 over the range $0.005 - 1.0\,\mathrm{m\,Pa^{-1}\,yr^{-1}}$. We find that for all but the smallest $C_0 = 0.005$, surging occurs and the dynamics of the events are as previously described. Example events are shown in Fig. 8. More detailed figures are contained in the supporting material.

Generally with larger $C_0$ the events are larger, losing a greater volume of ice from Hudson Strait, and both the peak and average speeds during the events are higher. The period and event duration does not appreciably change with the varying of $C_0$ suggesting that these time scales are set by the geometry of the ice sheet and/or the forcing fields.

## 4.2 Water Scheme

The water scheme used in all the preceding sections assumes that water produced in one grid cell is routed to adjoining cells in an approximation of a distributed drainage system (e.g. Creyts and Schoof, 2009). It is therefore still a question: are the surges the result of the particulars of the drainage scheme or can they arise in any system in which there is water at the base of the ice sheet and a sliding law that depends upon water depth?

To test this we shall describe in this section results for model runs using a very different water scheme: one which assumes that basal water is produced and dissipated locally beneath each individual grid cell. The water depth here is governed by Eq. 8. Where the previous water scheme represents a distributed drainage system, this one represents an efficient channelised drainage system where any water that is generated is quickly routed away, and crucially does not interact with the ice sheet in other regions.

We find that using this different water scheme, the ice sheet still displays periodic surges cycles. Figure 9 shows a similar time series to that in Fig. 3, using the same values of $C_0$ but the different water scheme.

The events are less regular than those using the Water Sheet Scheme, in both their timing and amplitude. The peak calving flux from the events is smaller, however, the length of the events is rather longer than those using the Water Sheet Scheme meaning that over the course of each event similar quantities of ice are lost from Hudson Strait.

The dynamics of the events with the new water scheme are similar to those described in Section 3.3, without of course the features that rely on the routing of water from one grid box to another. When the activation wave passes along Hudson Strait the water depth increases as it passes, with no delay. This makes for a faster initiation of the event. This can be seen comparing the green and black lines in the right panel of Figs. 3 and 9 which show the start of calving at the mouth of Hudson Strait and the time at which the centre of Hudson Bay registers the event with a decrease in ice sheet elevation. Using the Water Sheet Scheme there is a distinct 500 year gap between these events: using the Local Water Scheme they occur almost simultaneously. At the end to the event, using the Weertman water scheme there was a relatively rapid end to the event, the demise of the event was co-ordinated by water scheme. Using the new scheme the event takes much longer to end: compare the red and blue lines in Figs. 3 and 9. These show that the time between the surface elevation in the interior of the ice sheet stopping falling and the cessation of calving at the mouth of Hudson Strait is approximately 1000 years longer using the Local Water Scheme.

We also investigate the sensitivity of the surges to varying the parameter $C_0$ in Eq. 1 by again varying the parameter over the range $0.005 - 1.0\,\mathrm{m\,Pa^{-1}\,yr^{-1}}$. We find that the surging behaviour exists across this range, except for the highest value of $C_0$ (see Fig. 8 and Supplementary Information). Thus, we, once more, find that the surges are a quite robust feature of the model.





| Panel | $b$ (mm, half depth) | $a$ (mm, depth of transition) |
|---|---|---|
| (a) | 2.0 | 0.8 |
| (b) | 4.0 | 0.8 |
| (c) | 1.0 | 0.8 |
| (d) | 2.0 | 1.6 |
| (e) | 2.0 | 0.4 |
| (f) | 3.0 | 0.8 |
| (g) | 2.0 | 2.0 |
| (h) | 3.0 | 3.0 |
| (i) | 2.0 | 3.0 |

**Table 2.** Values of parameters used in the sensitivity test of Eq. 1 shown in Fig.10. Panels refer to the panel in Fig.10. $b$ and $a$ are the parameters in Eq. 1 and represent the depth at which the sliding parameter takes a value of $\frac{C_0}{2}$ and the depth over which the transition occurs, respectively.

This shows that surges are possible in our model regardless of the details of the way that water is routed beneath the ice sheet. The exact details of the surge are determined by the way that water is routed beneath the ice sheet but not their presence.

### 4.3 Water depth to sliding relationship

In Eq. 1 we relate the amount of sliding that occurs in the ice sheet to the depth of the water beneath the ice sheet. This is the crucial element of our surges as it determines the onset of sliding. We have seen in the previous sensitivity tests that the surging behaviour is quite robust to changing other elements in the model. In this section we investigate how robust the surges are to changes in the switch.

Equation 1 contains two key parameters: how much water must accumulate at the base of the ice sheet for sliding to occur (parameter $b$) and over how much water depth the transition from a small fraction of the sliding to full sliding occurs (parameter $a$). There is no *a priori* values that these parameters should take therefore in this section we vary both of them to investigate what range they may take. On the low side, we are constrained by the limit that sliding may not occur for water depths less than or equal to zero, therefore no set of parameters allow this to occur. On the upper side we raise the value of $b$ until we no longer find that the ice sheet model surges over Hudson Bay. A summary of the values we investigate are shown in Table 2.

Figure 10 shows the thickness of the ice sheet over Hudson Bay (the region shown in Fig. 3) for 50kyr after a 50kyr spin up of the ice sheet model from the ice5g configuration. We find that the surging behaviour stops when half the sliding occurs at a water depth greater than 4 mm. Below this depth surging still occurs. With progressively less abrupt transitions the surging behaviour still occurs although there is a tendency for shorter and more frequent events with less abrupt transitions.

Again we find that the surges can occur over a wide range of parameter values suggesting that they are a robust physical feature of the model.





## 4.4 Resolution

One common feature of previous studies examining surging events in realistic three dimensional ice sheet models is the very narrow width of the ice streams in Hudson Strait. Although there is evidence for the narrowness of the ice streams associated with HE surges (Stokes and Tarasov, 2010), in many models the ice streams width is one or two grid points. This suggests that

the model ice streams may be manifestation of numerical instabilities within the models (Hindmarsh, 2009). For example Papa et al. (2005) report that their model is very close to numerical instability. Similarly many of the ice sheet models used in the HEINO intercomparison (Calov et al., 2010), show a curious structure to the surging behaviour that bears many hallmarks of numerical, rather than physical instability mechanisms.

One way to investigate whether the surges are a numerical artefact is to use different model resolutions. Higher resolution

runs can better resolve the ice streams and, if we find that the structure of the surges is the same regardless of resolution it is likely that they are not the result of the numerics of the model. If we find, however, that the ice streams remain only a few grid points wide we must seriously consider whether these ice streams and HE are indeed physical. In this section we carry out such tests, varying the resolution of our model to assess how the surging events change.

We run the model at progressively finer resolutions, (50km, 30km and 25km) and find surging behavior at all three resolu-

tions. The behavior of the events are broadly similar, with events being of similar size and duration. This is strongly indicative of the robustness of the events to resolution. We did not run the model at higher resolution as it was computationally prohibitive.

We see in Fig. 11 that the width of the ice stream along Hudson Strait is the same regardless of the resolution, and takes up the entire width of the fast sliding region. The structure of the surge within Hudson Bay itself is more complicated, this is likely due to the more detailed bottom topography that the higher resolutions allow.

The striking similarity between the structure of the surges is indicative that the surges are not a numerical artefact but rather they are a physically based process.

## 5   Discussion

We have shown that it is possible to get binge-purge cycles in the Glimmer ice sheet model. The size and period of these events varies with the choice of parameters and parameterizations. We shall now compare these results with previous studies and

observations. Table 1 summarizes our results and those from a number of previous modelling studies as well as estimates for the vital statistics of HE from observations.

The first comment that must be made is that the size and duration of HE is very uncertain. There is no direct way of measuring the size of HE from observations, it can only be estimated using suitable assumptions. Similarly their timing and duration is very hard to estimate since by their nature they are highly anomalous events and thus do not conform to age models derived

from more normal conditions.

That being said, the duration of the events that we model are long compared to the observations, taking on order 3000 years rather than the observed $495 \pm 255$ years (Hemming, 2004). They also last longer than events in many of the other ice sheet models, although they are consistent with the more realistic, three dimensional ice sheet models (Álvarez Solas et al., 2011).





This suggests that the model configuration presented here, and indeed other similar ice sheet models, may be in the sliding regime for too long. In our model the excessive length of the HE could be due to the water drainage scheme allowing water to remain beneath the ice sheet too long. Fowler and Schiavi (1998) found that in their model longer water relaxation times, that is the time taken for water to drain from beneath the ice sheet, gave rise to longer lived events. We are not aware of the
reason for the too long event duration in other models. We stress that this problem is a common feature of ice sheet models that simulate surges of ice from Hudson Bay. It occurs regardless of the trigger for the surge.

The events that we simulate are larger than observational estimates suggest (although the exact size of the events is highly uncertain). This is likely to be associated with the excessive duration of the events that we simulate: given the same flux from the ice sheet a longer event will give you a larger net loss of ice. Interestingly the flux of ice that the surges give rise to is one
feature that is remarkably consistent among ice sheet models.

All of the ice sheet models suggest that the peak flux of ice from Hudson Bay is around 0.05 Sv, and certainly less than 0.1 Sv. This is consistent among ice sheet models with quite different numerical formulations (e.g. the SIA formulation in this study and the hybird SIA/shallow shelf formulation used by Álvarez Solas et al. (2011)). This is not altogether surprising because the flux is set by the geometry of the ice sheet, in particular the width of Hudson Strait and the maximum speed that ice
can slide and we would not expect either of these to be hugely different from model to model. This consistency among models has its own implications: if the ice sheet models are correct, simulating the effect of HE with arbitrary freshwater fluxes that exceed these values (as is typical e.g. Kageyama et al., 2013) is inconsistent.

The interval between events in our models is somewhat longer than observed, occurring every 11,000 years, with a range of 10,000 to 21,000 for other configurations. These periods are somewhat longer than the observed period of around 7000 years.
There are number of possible reasons for this. It is most likely the result of the constant climate forcing that we use. Payne (1995) showed that the accumulation rate over the ice sheet plays a pivotal role in determining the return period of events, with more accumulation giving more frequent events. Computing the surface mass balance field using simulations of 40ka using the model HadCM3 Singarayer and Valdes (2010), a climate model related to FAMOUS, we estimate that the surface mass balance is up to twice the size of that in a 21ka simulation using the same model. This suggests that we are severely underestimating
the accumulation rate in our LGM simulation. A larger accumulation rate would bring the period of the surges far closer to the observed 7ka. Furthermore, we assume a constant forcing to the ice sheet, where in fact the ice sheet and climate are coupled, hence the surface mass balance changes in response to the evolving ice sheet topography. Interestingly coupled simulations using the same ice sheet configuration but with an interactive climate model do not show a significant change to the timing of the HE (Roberts et al., 2014b). However, in both the coupled simulations and those presented here we use an LGM climate.
The LGM was the coldest period during the last glacial, however, HE occurred throughout the last glacial period at times when temperatures were warmer than at the LGM. Warmer surface temperatures would lead to a warmer ice sheet whose base would take less time to melt, and, furthermore, in a warmer climate we might expect higher accumulation rates. Finally, there is the possibility that the occurrence of the HE is paced by external forcing.

As we outlined in the Introduction there is considerable debate at present as to whether HE are triggered by some external
forcing. The results of our study can neither confirm nor refute this claim. As described above our simulations do overestimate





the period of the events and this is likely due to our using an LGM climate to force the model. However, our results do show that when the hydrology at the base of the ice sheet is better simulated than in previous studies, ice sheet surges can still occur. The surging behaviour is not a numerical artefact of the model. Therefore the binge-purge mechanism can not be rejected as an explanation for HE on the basis that is a modelling quirk, just as it can not be rejected on the basis of the available data.

## 5    6    Conclusions

We have shown that it is possible to simulate binge-purge events in the Glimmer ice sheet model using basal water depth as a trigger mechanism. These events have a period that is close to the observed period of Heinrich Events and the size of these events is also within the range of observed events.

These events arise from the ice sheet switching between two states: slowly creeping when the base is frozen to the bed and
quickly sliding when the base is sitting on a sheet of water. In contrast to previous studies, the switch between these two states is simulated using a *tanh* function of water depth that allows for a smooth transition between sliding and creeping over a range of water depths.

The occurrence of the events is the result of a slow warming at the base of the ice sheet that gradually brings the ice sheet bed to pressure melting point, at which time a layer of water can form at the base of the ice sheet. This warming is the result of
the geothermal heat flux and, especially in the Hudson Strait region, the strain heating. When the ice sheet bed is at pressure melting point, a water sheet can form beneath the ice sheet, but this sheet is not necessarily thick enough to allow fast sliding. For example, in Hudson Bay, the ice sheet base is at pressure melting point for many years before a surge because the water sheet thickness beneath the ice is not very thick. What is required to deepen the water at the base of the ice sheet is an increase in heating rate, this arises from strain heating. When the activation wave of strain heating passes through the ice sheet the
water depths are deepened enough to allow fast sliding. Once the ice sheet is sliding fast, frictional heating can maintain the water depth at sufficient depth to allow sliding to continue. The surge ends when the gradient of geopotential diminishes. This reduces the driving stress, which in turn reduces the speed of sliding, the heat generated by friction and the water depth. The reduction in the gradient of geopotential also reduces the depth of water beneath the ice sheet directly by routing less water from the interior of the ice sheet to the margins.

These surges occur with a period of between 10 and 21 thousand years. Because we use a constant climate forcing we can say that the surges are an intrinsic feature of the ice sheet. The peak flux of ice out of the mouth of Hudson Strait is 0.05 Sv, with a flux of 0.06 Sv possible using a particularly fast sliding ice sheet. The size of the surges is $2.50 \times 10^6$ km$^3$ with a maximum of $4.30 \times 10^6$ km$^3$ and minimum of $1.90 \times 10^6$ km$^3$ possible when different values of the sliding parameter are used. We must though note that the very largest events have an unrealistically long period compared to the observations. These statistics are
very similar to other modelling studies and lie within range of the observations. We should note, however, that the time that it takes for the surges to occur is longer than the observations suggest.

We vary a number of parameterizations and parameter values within the model to test how sensitive the model is to values of these parameters. We find that in all but a very few cases periodic surges occur. This robustness suggests that the surges are not



the result of a particuliar set of conditions or a numerical instability in the model. Tests that vary the resolution in the model are especially indicative that the surges are not the result of a numerical instability, since the gross statistics of the events are very similar for each of the progressively finer resolutions examined.

This shows that the binge-purge mechanism can operate in a complex 3-D ice sheet model which incorporates realistic geometry from the last ice age over the Laurentide ice sheet. Furthermore, because the binge-purge cycles that the model simulates compare well with the bulk of the observations of Heinrich Events we can not only say that binge-purge cycles could exist in Laurentide Ice Sheet but also that these cycles could explain Heinrich Events.

*Author contributions.* WHGR, AJP, and PJV designed the research and wrote the manuscript. WHGR completed the numerical simulations

*Acknowledgements.* This research was made possible through NERC grants NE/I010920/1 and NE/G006989/1. Computer simulations were carried out using the computational facilities of the Advanced Computing Research Centre, University of Bristol - http://www.bris.ac.uk/acrc/.



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





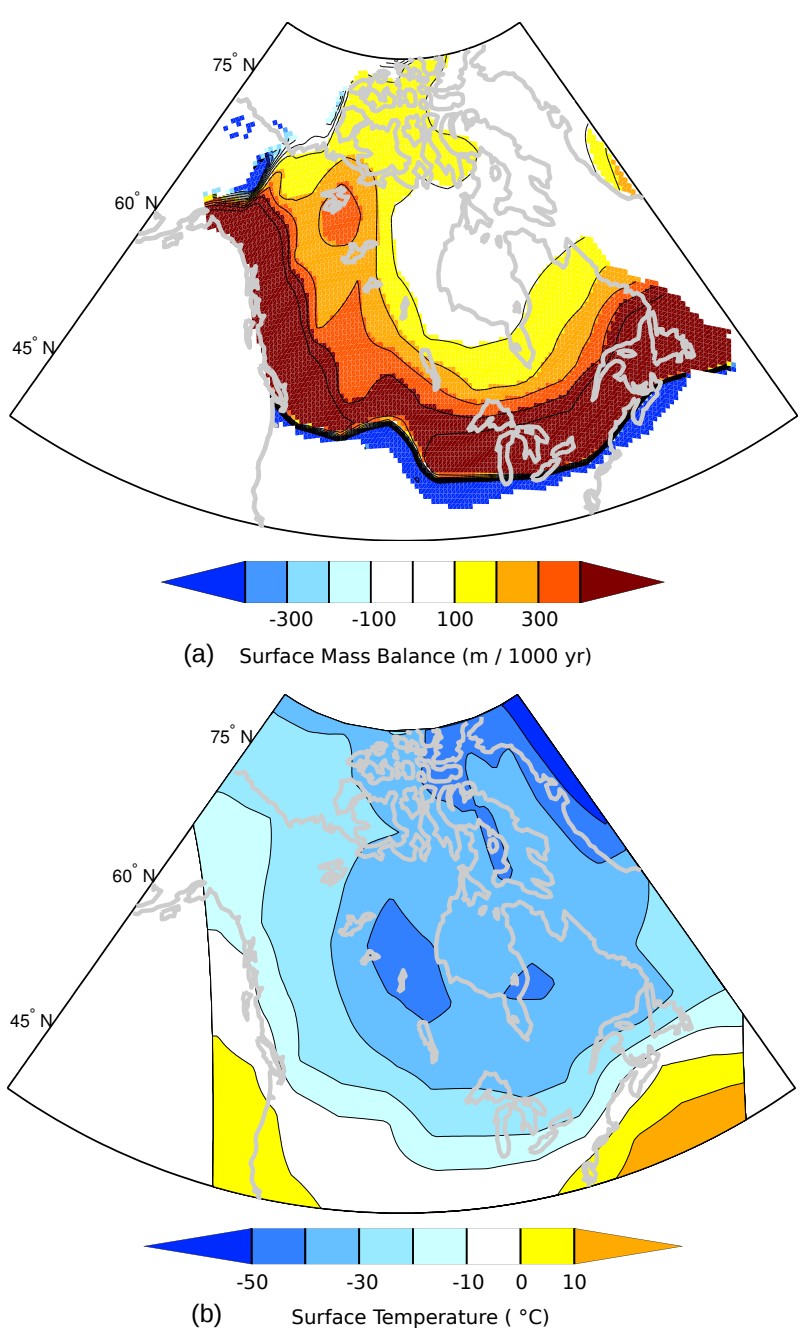

**Figure 1.** Ice sheet forcing fields. (a) shows the annual average surface mass balance field and (b) the annual average surface temperature field





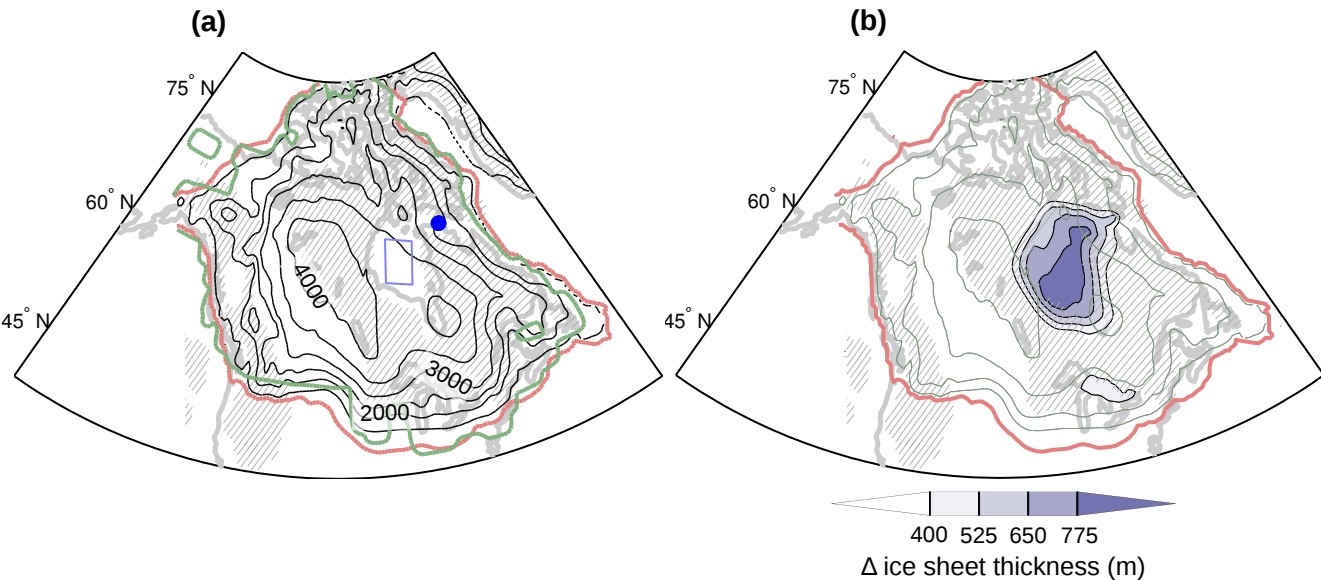

**Figure 2.** The mean height of the ice sheet (a) and the mean change in topography over a composite Heinrich Event (b). In (a) the black contours show the height of the ice sheet, contour interval 500m. The red contour shows the extent of the ice sheet as modelled by Glimmer, the green contour shows the extent of the ice5g ice sheet (Peltier, 2004). Hatched areas indicate areas of hard bed, where the sliding parameter is very low, the remaining areas are soft bed with higher sliding parameter. The blue square is the region over which the upper curve in Fig. 3 is averaged. The blue dot shows the location from which the time series in Fig. 5 are derived. In (b) we show the composite difference between the ice sheet thickness before and after a HE. The composite is the average over all HE that occur during the 100kyr run of the model shown in Fig. 3.





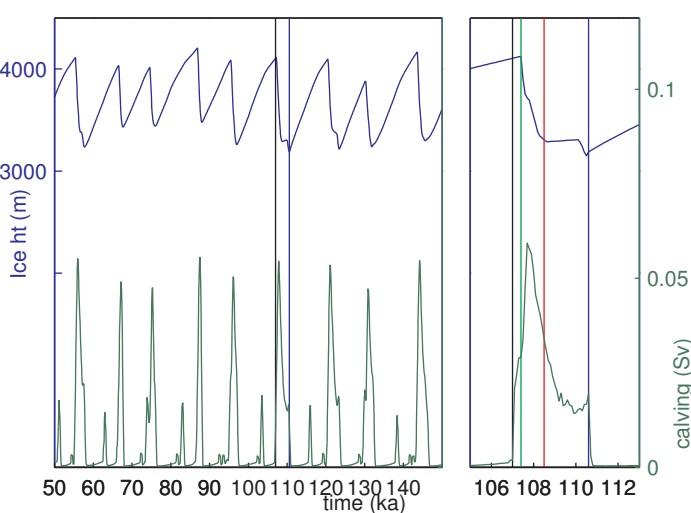

**Figure 3.** Variability in the ice sheet over Husdon Bay and Hudson Strait. The upper curve shows the average height of the ice sheet over the central Hudson Bay (m, blue box in Fig. 2), the lower curve is the flux of ice out of the mouth of Husdon Strait (1 Sv = $1 \times 10^6 \mathrm{m}^3 \mathrm{s}^{-1}$). The left panel shows results for 100,000 years of model run, the right panel zooms in on the event between 105-113 kyr.





**Figure 4.** Time slices of the basal temperature field taken at (a) 27000yr, (b) 27500 yr, (c) 28000 yr, (d) 28200 yr. The times shown are the same as those in Fig. 5. Temperatures are plotted relative to the pressure melting point, with red colours indicating where the ice sheet base is at pressure melting




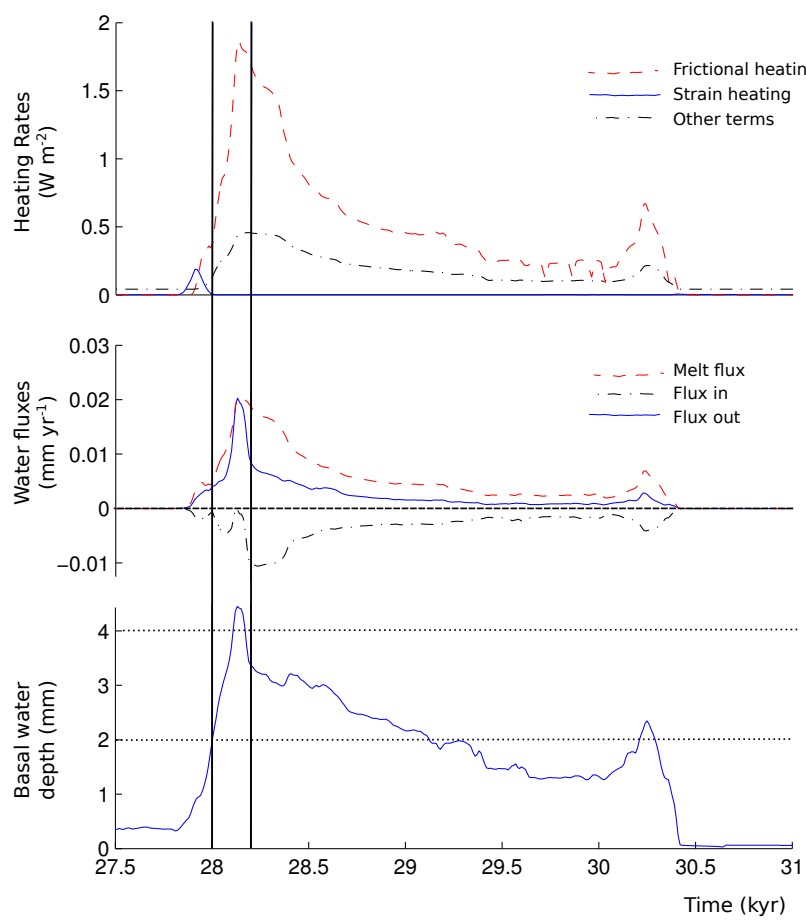

**Figure 5.** Time series of various quantities averaged across Hudson Strait at the location marked by the blue dot in Fig 2(a). The top panel shows heating rates in the lowest model level (W m$^{-2}$). Red dashed line, frictional heating; blue solid line, strain heating; black dashed line, other heating terms. The middle panel shows water fluxes (mm yr$^1$). Red dashed line, melting; blue solid line, flux of water out of the box, the flux from which the water depth is calculated; black dashed line, the difference between the flux out of the box and the melt rate. A positive value means that there is a flux of water from upstream into the box, a negative value means that the flux of water out of the box is greater than the melt rate and the box loses water down stream. The bottom panel shows the water depth at the base of the ice sheet (mm).





**Figure 6.** Time slices of the basal water depth field taken at (a) 27000yr, (b) 27500 yr, (c) 28000 yr, (d) 28200 yr. The times shown are the same as those in Fig. 5. When water depths reach 2mm, the maximum shading, the sliding parameter $C$ takes a value of $0.5\,C_o$. The contours show selected values of the gradient of the geopotential in order to show the passage of the activation wave



**Figure 7.** Time slices of the vertically averaged velocity field taken at (a) 27000yr, (b) 27500 yr, (c) 28000 yr, (d) 28200 yr. The times shown are the same as those in Fig. 5. Note that the colours are plotted on a logarithmic scale.





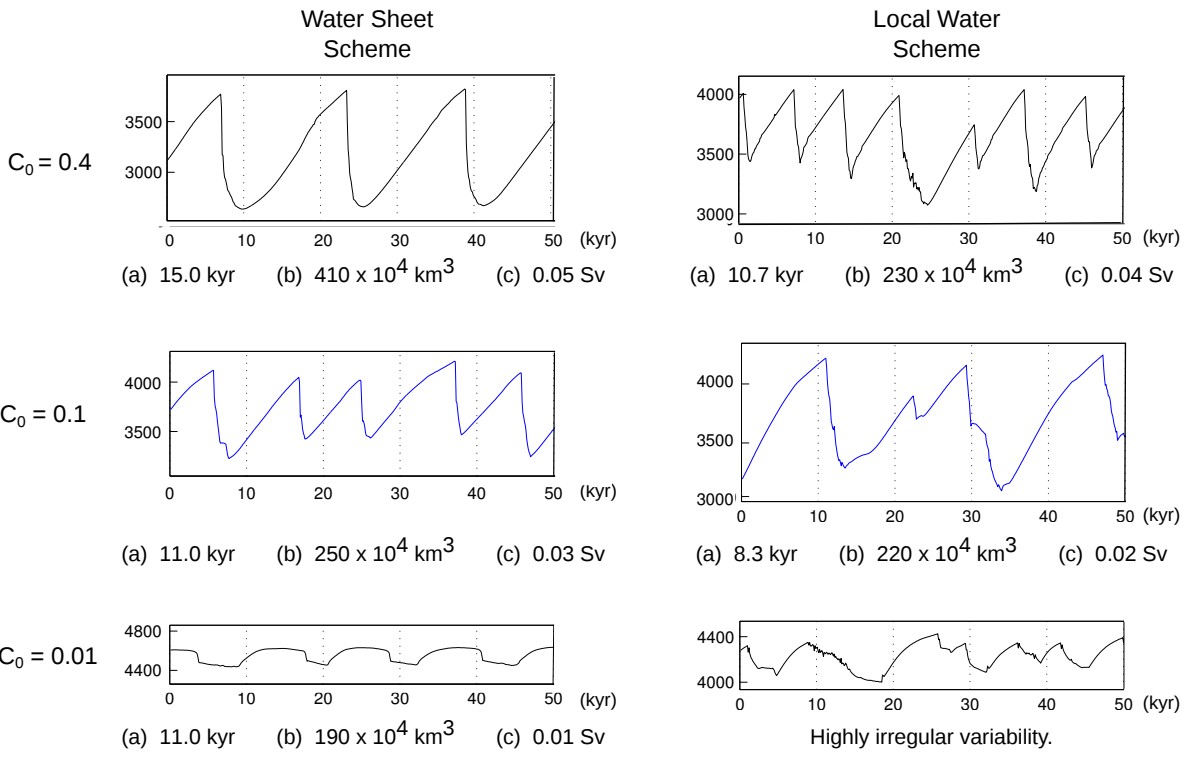

**Figure 8.** Summary of sensitivity to sliding parameter. We show the height of the ice sheet in the central Hudson Bay for 3 different values of the sliding parameter and for the two different water schemes. Example statisitcs for typical events, averaged over the full 100 kyr are also shown, these are: (a) the period, (b) the size, (c) the peak calving rate





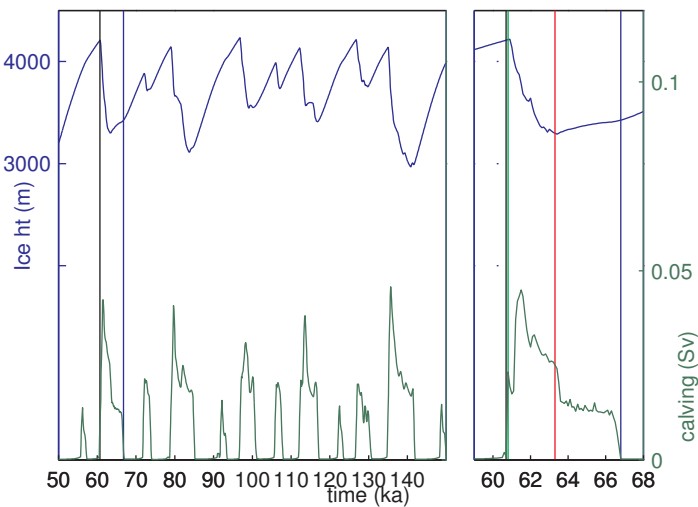

**Figure 9.** Variability in the ice sheet over Husdon Bay and Hudson Strait using the Local Water Scheme. The upper curve shows the average height of the ice sheet over the central Hudson Bay (m, blue box in Fig. 2), the lower curve is the flux of ice out of the mouth of Husdon Strait (1 Sv = $1 \times 10^6 \, \mathrm{m}^3 \, \mathrm{s}^{-1}$). The left panel shows results for 100,000 years of model run, the right panel zooms in on the event between 60-68 kyr.





**Figure 10.** Sensitivity to the parameters (*a*) and (*b*) in Eq. 1. Theses values are presented in Table 2. Panels (a)-(i) show the surface height of the ice sheet averaged over the blue box in Fig 3. Panel (j) shows the sliding fraction, $\frac{C}{C_o}$ from Eq. 1, as a function of water depth, for the different values.





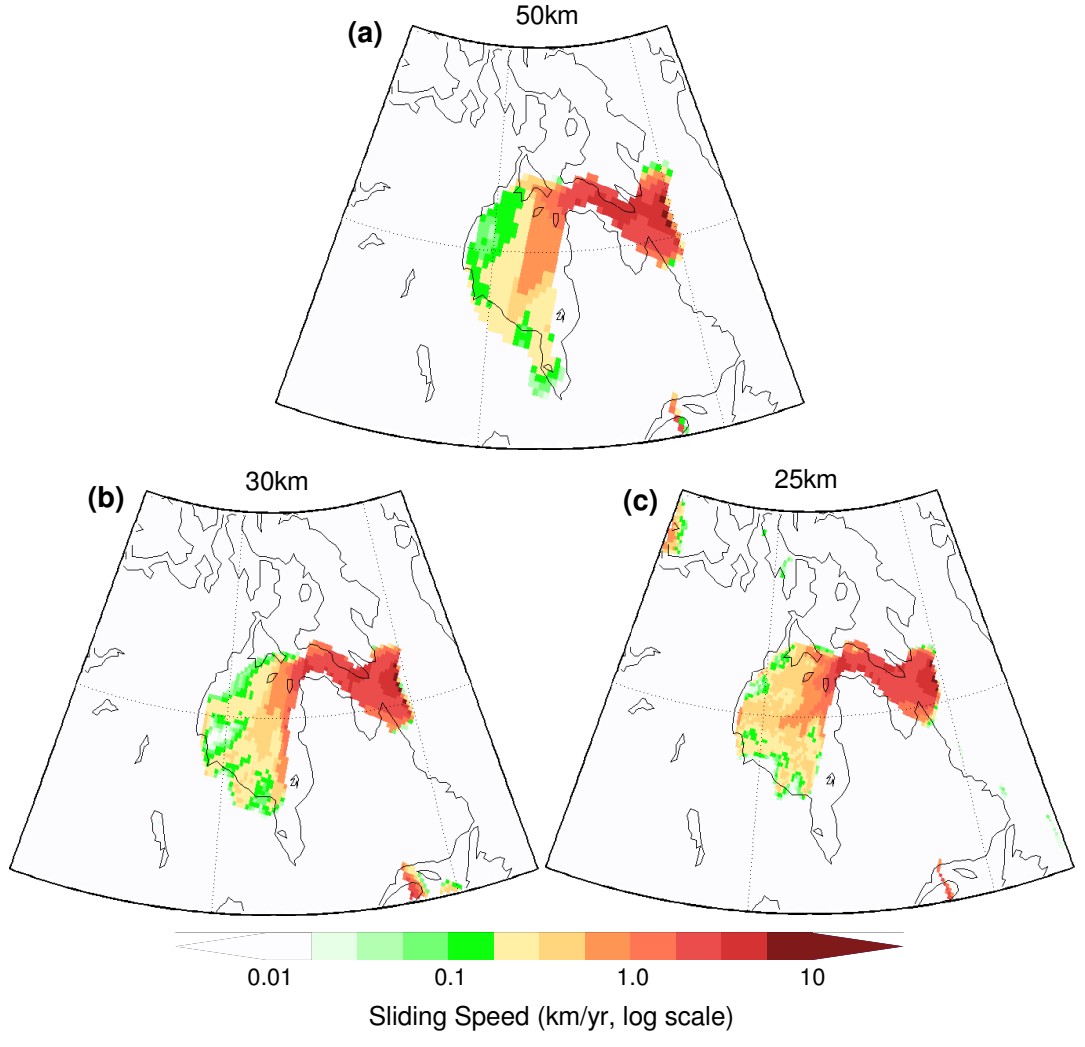

**Figure 11.** Instantaneous basal sliding speed for model runs at 3 different resolutions, (a) 50km, (b) 30 km, (c) 25km. Colours for speed are plotted on a logarithmic scale.