# Peer review of "The role of basal hydrology in the surging of the Laurentide Ice Sheet"

_Climate of the Past, 2016_

## Referee Comment (RC1) · L. Tarasov (Referee) · 15 Mar 2016

This submission examines the possible role of basal hydrology in Hudson Strait ice stream discharge cycling (aka Heinrich Event source). The dynamical source of Heinrich Events is a major ongoing question for the paleo community as well as a challenge for the modelling community.

Though the duration and periodicity of the Hudson Stream surging are somewhat off, the surge cycling modelled in this submission is otherwise of reasonably magnitude. A number of relevant model uncertainties are examined via sensitivity tests: parametric (sliding speed scaling parameter and dependencies on basal water depth), and to a lesser extent, model resolution. The results show the response to be relatively robust which is a significant modelling accomplishment. The paper is overall wellwritten/edited (I did not find any typos), with generally appropriate figures and tables and supplement. There is some thoughtful analysis ("Anatomy ...") examining the details of surge activation and deactivation.

It would have been nice to see a 10km resolution test, as a factor 2 total range for the sensitivity test is quite limited. Also, there needs to be corresponding time-series and table entries for resolution response as is provided for the parametric sensitivity tests.

The study did leave me with two outstanding questions. At the beginning, it reiterates that standard understanding that the shallow ice approximation (SIA) for ice dynamics is inappropriate for ice streams but then ignores this throughout the remainder of the text. There should at least be some discussion of what the potential impacts would be from including a higher order representation, or hybrid/membrane approximation for the ice dynamics (eg with reference to the MISMIP comparisons...). The study obtains Heinrich type cycling with a model that uses a representation of ice streaming that is not physically defensible. The relative robustness of the cycling response to tested model parameters and even to the complexity of the hydrology model offers some confidence in the model. But with confirmation from an appropriate ice dynamics representation, there would be significantly more confidence in the physical validity of the results.

Glimmer has higher order physics options. Could this not be turned on for a short (eg 15 kyr) test? PISM with membrane approximation could be run at 25 km resolution over a glacial cycle as could Dave Pollard's hybrid model. I know this is non-trivial, and therefore would not use this as grounds for rejection of this submission but I urge some exploration of time feasibility of this.

The other issue is the choice of water depth dependence as opposed to basal water pressure. There is no clear physical model of why a 2-4mm basal water depth would be a reasonable threshold for fast basal sliding, especially consider actually topography/surface roughness at say 1 m scale... Computationally, water depth offers an easier to implement model and I can speculate on some physical motivation. There is

some discussion of this in Le Brocq et al., 2009, and it would help this paper to provide further justification.

The final discussions/conclusions would also benefit from more attention to remaining uncertainties and implicit assumptions. What impact might higher order physics have on the surge cycling duration and periodicity?

**Specific comments**

The depth of water beneath ice sheets has been argued to be intimately related to the speed with the overlying ice can slide (Budd and Jenssen, 1987; Le Brocq et al., 2009).

**Yes but the 2nd reference also raises the issue of how to reconcile mm scale water depth with potentially metre scale water storage in subglacial sediment. It needs to be made clear that this parametrization as of yet has no clear physical basis.**

We assume here that the effective pressure is zero (see, e.g., Budd and Jenssen, 1987; Alley, 1996). Although we would expect the effective pressure to have an impact upon the rate of sliding we neglect this effect as it is small.

**I see no basis for either of the above claims (depending what "close" means, presumably small enough to be ignored) given current literature and understanding (eg Cuffey and Patterson, 2010, for a broad review).**

If temperatures are anomalously cold we would expect a reduction in the mass lost from the ice sheet from surface melt but an increase in the mass lost due to calving.

**The later does not follow necessarily. Perennial landfast sea ice could choke up the system as presently observed seasonally for tidewater glaciers in the Arctic. Cold conditions could also reduce thermal forcings of calving.**

An increase in the calving could make it easier for the freshwater from the ice sheet to impact the AMOC, but it will undoubtedly also increase the ice shelf's thickness making

it more resistant to melt and a better buttress.

**Do you mean thickness at the calving front? I would expect thickness at the grounding line to decrease with increasing calving (with some time lag) due to less buttressing from less shelf extent**

We acknowledge this omission but must neglect it since using higher order approximations make the long model integrations that we need to perform computationally impossible.

**Could you at least do a 40 kyr integration at 10 km, interpolating a restart file from the 25 km run to avoid the spin-up?**

At the base of the ice sheet the vertical gradient of temperature, contained in the vertical advection and diffusion terms is a result of heating by the geothermal heat flux and heating due to friction at the bed.

This warming is the result of the geothermal heat flux and, especially in the Hudson Strait region, the strain heating

**Incorrect. Basal temperature is the result of energy conservation, and is therefore due to all terms. Your figure 5 shows that "other terms" contributes more than strain heating.**

Previous models have taken as the switch the temperature at the bed of the ice sheet (Calov et al., 2002, 2010; Papa et al., 2005).

**Not all models, eg Johnson and Fastook, 2002**

The behaviour of the events are broadly similar, with events being of similar size and duration. This is strongly indicative of the robustness of the events to resolution.

**"similar size and duration" with presentation of the actual results does not provide any evidence of robustness. Provide a time series comparison.**

This compares well with the ice5g distribution that the model was initialized with, which has an area of 1.68×107 km2.

**Not surprising, if ice5g was used as the boundary condition for the FAMOUS run**

At some time the base of the ice sheet will warm sufficiently that the gradient in ice sheet surface, and its associated strain heating, can warm the interior of the ice sheet above pressure melting point.

**Is the basal water flow blockage switch off/on at the pressure melting point? This would not be physical as a 50 km square block of ice won't freeze or get warm-based simultaneously across its base and the experiment should be repeated again with a smooth ramped transition over some range O(0.1 to 0.5 K)**

As the water depth increases the sliding speed increases and thus the heating rate from friction can increase.

**Physically, increasing water depth decreases effective basal drag to permit increased sliding speed, so its not clear if the heating rate from increasing water depth should necessarily increase though it's clear why it does in the current model.**

These two regions are determined using a global sediment thickness map (Laske and Masters, 1997).

**Caveat, this thickness map was created for a seismology context and has numerous errors for a glaciology context.**

reasonably simulate sliding at the base of the present day West Antarctic Ice Sheet

**vague claims such as the above are common within the ice sheet modelling community, but indefensible. Be more precise.**

**FIGURES:**

**figure 10, need to label plots (a=, b=) so that the reader can decipher without opening up another page**

[Figure]

---

## Referee Comment (RC2) · Anonymous Referee #2 · 15 Mar 2016

**Review of: *The role of basal hydrology in the surging of the Laurentide Ice Sheet**

by William H. G. Roberts, Antony J. Payne, and Paul J. Valdes

**General Impression**

This article uses a model setup that - how I understand  it – has been already applied to the West Antarctic ice sheet (LeBroq et al., 2009) and now has been adapted to investigate ice-sheet instabilities of the Laurentide ice sheet in the context of Heinrich events. The article is well structured and the necessary details to understand the model assumptions are clearly laid out. Figures and tables

I am aware that in the case of dealing with simulations of such long time scales, the choice of models is limited by the needed approximations to keep the computational costs in a manageable range. Nevertheless, I think you picked the wrong approximation.  I have the suspicion that a SIA model is an over-simplification to the problem, simply by the undeniable fact that it is based on assumptions that are opposite to the physics of fast ice outlets and that a model accounting for longitudinal stresses would have been the correct choice. I elaborate these points of criticism in the following section.

I think, that in the end it is about judging the relative importance of the (apparently to the mechanical problem of fast flow wrong) ice mechanics as well as (to a lower degree of importance) the displacement of the lithosphere under the load signal in relation to the hydrology model (which itself would demand a deeper discussion) in this coupled system. If the authors can come up with an improved chain of arguments justifying the combination of partly severe model assumptions, I see no problem of having this article published. The easiest way to show it would be to include higher order effects (i.e., use an improved ice-flow model) and compare to this solution.

**Major points of critics**

My major concern is about the choice of the ice sheet model used in this investigation. You use a model that is based on the shallow ice approximation (SIA). SIA basically only is capable of correctly representing flow situations linked to creep flow with little to no sliding in absence of longitudinal and transversal stresses – a situation completely opposite to ice streams. Throughout the text you frequently correctly point out the issues linked to this approximation in connection to fast flow features including citation of the findings by Hindmarsh (2009) that summarize those concerns. On **page 5** starting from **line 13** you state:

> The SIA neglects longitudinal stress gradients. Although these stresses are negligible in the interior of a slow moving ice sheet, they are important at the margins where they are integral to ice shelf and ice stream dynamics. Furthermore, in regions where horizontal shearing is important, for example at the boundary between slow moving ice and fast moving ice streams, longitudinal stresses are not negligible (Hindmarsh, 2009). The lack of longitudinal stresses in regions of high horizontal shear is of concern since we would expect such areas of high shear to occur during surging events when parts of the ice sheet are moving at relatively high velocities whilst surrounded by areas of much more slowly moving ice. We acknowledge this omission but must neglect it since using higher order approximations make the long model integrations that we need to perform computationally impossible.

I do not agree on the justification in the last sentence of this paragraph. Adopting the common

nomenclature of the expression "higher order model" (="anything else exceeding SIA in complexity") in literature, there are examples that contradict your statement. There are known studies stretching over at least similar time-scales using hybrid models that swap the SIA with the shallow shelf approximation (SSA) in areas with dominating horizontal stresses. The latter being a better approximation (using a still on shallowness based assumption of plug-like ice flow) to streaming ice flow that provides a computationally relatively low-cost implementation. Besides the application using GRISLI cited within the article by yourself (Álvarez-Solas et al., 2011), I instantly could mention Pollard and DeConto (2013) who applied such a model (including a simplified model for grounding line migration) to the Antarctic ice sheet and the article by Bindschadler et al. (2013) which contains long-time integration of shelfy-stream models for the Greenland ice sheet. Even a spin-off from Glimmer, Glimmer-CISM (Lipscomb et al., 2013), would have the ability to better represent the dynamics of fast flow features in the ice sheet. Also L1L2-type of models, such as BISICLES (Cornford et al., 2013), might be able to handle simulations of such times spans. In short: **I not to all extend understand the choice of the model (in terms of physics), and have troubles to accept the justification for doing so**.

The second topic that would need elaboration is the **physical concept of the hydrology model** and not restricting yourself to studies on the parameters within. Is the solution for laminar flow between two plates an adequate description for the water transfer through aquifers in sediments? This part at least demands some discussion.

Lastly, but this might not be a big issue after all, but at least demands clarification: On **page 8, line 15** you state (mind also the typo: *…, we use …*):

> Although Glimmer does allow for the use of a lithosphere model beneath the ice sheet, in order that we can make direct comparisons between the different runs in the suite of sensitivity tests, we uses a topography beneath the ice sheet that does not vary in time. For this we use the ice5g topography for 21ka (Peltier, 2004)

If you have the ability to account for **isostasy** (presumably this is what you meant by lithosphere model), **why not taking it in?** If you have something like a LDRA model at hand, then this should be relatively cheap to include such a run in the sensitivity analysis to exclude any influence of an assumed fixed bedrock. Despite the fact that by Shreve's assumption the gradient of the ice-sheet surface has a to the bedrock gradient by an order-of-magnitude stronger influence on the hydro-potential, which manifests itself in (the corrected version of) equation (5), it would be beneficial to explicitly mention (perhaps supported by a reference) that, compared to changes in ice surface gradients, changes in bedrock elevation gradients due to changes in ice-load over the whole simulation area as well as time-period have no significant influence on hydrology. In particular as you write yourself about the sensitivity of surges with respect to bedrock resolution (**page 15, line 18**:)

> up the entire width of the fast sliding region. The structure of the surge within Hudson Bay itself is more complicated, this is likely due to the more detailed bottom topography that the higher resolutions allow.

To give an example where neglecting lithosphere dynamics might be an issue: Bedrock gradients can affect if they apply in regions with $\nabla \Phi \approx 0$, i.e., over regions of low surface slope, which usually coincide with regions of largest ice thickness and hence largest bedrock displacements (and consequently largest changes in displacement) to decide on the principal direction of the water routing (I recall on the findings of AGAP around Dome A in Antarctica!).

**Minor issues**

**Page 7, equation (5):**

$$\nabla\Phi = \rho_i g \nabla S + (\rho_w + \rho_i) g \nabla h. \tag{5}$$

I think the brackets should contain the difference rather than the sum of the densities, hence $(\rho_w - \rho_i)$ as else the bedrock influence on the hydro-potential would be about twice the one of the free ice-surface – in reality it is 1/10-th.

**Page 8, line 21:**

is made. Diagnostic fields are output every 100 years. However when diagnosing the processes responsible for the surges we use output derived every time step.

The last sentence would need a definition of your time-step size, for convenience in the same paragraph.

**Page 8, line 24:**

the FAMOUS climate model. The surface mass balance used by the ice sheet model is calculated using the precipitation and temperature fields from the climate model and use a simple positive degree day scheme (Reeh, 1991; Rutt et al., 2009). The

If the resolution of your ice sheet does not coincide with the climate model, please drop a line on how you interpolate/downscale your forcing fields. Do you account for elevation lapse rates?

**Page 8, line 27:**

also shown, in Fig. 1(b). The base of the ice sheet is forced with a spatially and temporally constant geothermal heat flux that takes a value of $4.2\times10^{-2} Wm^{-2}$.

The choice of a spatial and temporal constant numeric value of the heat flux deserves a justification.

**Page 8, line 31:**

tion uses the Water Sheet Scheme, a sediment sliding parameter of 0.1 m $Pa^{-1}$ $yr^{-1}$, and a hard bed sliding parameter of 0.005 m $Pa^{-1}$ $yr^{-1}$. Firstly we describe the mean state of the ice sheet.

Same as before. Do these value somehow link to something in literature or are they tuned parameters? Having read further, I conclude they might be chosen due to findings from other studies – perhaps mention that already here.

**Page 8, missing information:**

You are not revealing details on the boundary **condition** imposed at the **ice-sheet/ocean boundary**. This is in particular of interest as you report on different calving fluxes in the discussion later on. What does "calving flux" in terms of your setup mean? The two options you have with SIA is fixed horizontal boundary (and calving is equal to the ice flux through this boundary) or that you allow to advance the ice sheet along the sea-floor, describing ice loss (a.k.a. calving) below a certain depth. In view of some of the HE theories

involving ice-shelf dynamics, I would ask to have the consequences imposed by the ocean boundary of the ice-sheet model discussed (missing buttressing, no grounding line migration, etc.).

**Page 15, sub-section 4.4**:

> The striking similarity between the structure of the surges is indicative that the surges are not a numerical artefact .

This very much links to the major point of criticism on model choice. At the best you can say that by your series of runs presented in Fig. 11 you give indication of **numerical convergence** of your setup. Still, this applies only on the level of your discretized model, which might or might not represent physics. Hence, I suggest to drop the last part of the sentence. If you could compare the results of Fig. 11 to a reference solution (e.g., from another model with a suitable mechanical representation of ice flow, such as a SSA model), this then could give some indication on physics. **For the moment all you can claim is that you have indications (not even the proof) to have a converging numerical model that is capable of producing cyclic behaviour, but nothing beyond that.**

**Typos, etc.**

**Page 1, line 5:** the period is missing at the end of the sentence:

> contrast to many previous studies where surges only occur for rather specific cases. The robustness of the surges is likely due to the use of water as the switch mechanism for sliding. The statistics of the binge-purge cycles resemble observed Heinrich

**Page 2, line 33:**

> if the ice sheet must be forced we need to find another external trigger mechanism. In particular, such external forcing would require ice shelves to exist in other sectors of the ice sheet, areas which may be less conducive to their formation that the Labrador Sea.

than?

**Page 4, line28** (and on different other places):

> bed being kept at pressure melting. Thus, although 3D ice sheet models have been coupled to basal hydrology models, their configuration has never been realistic enough to show the importance of water in ice sheet surges.
>     In this study we shall simulate HEs in a 3-D ice sheet model, Glimmer (Rutt et al., 2009), using such a water scheme. We

Very minor issue, but I suggest to consistently use either 3-D or 3D.

**References**

Bindschadler, R. and 27 others, 2013: Ice-sheet model sensitivies to environmental forcing and their use in projecting future sea level (the SeaRISE project). J. Glaciol. 59, 195-224,  doi: 10.3189/2013JoG12J125.

Cornford S, Martin D, Graves D, Ranken D, Le Brocq A, Gladstone R, Payne A, Ng E and Lipscomb W,  2013: Adaptive mesh, finite volume modeling of marine ice sheets. J. Comput. Phys. 232, 529–549, doi:10.1016/j.jcp.2012.08.037

Lipscomb, W. H., J. G. Fyke, M. Vizcaíno, W. J. Sacks, J. Wolfe, M. Vertenstein, A. Craig, E. Kluzek, and D. M. Lawrence, 2013: Implementation and Initial Evaluation of the Glimmer Community Ice Sheet Model in the Community Earth System Model. J. Climate, 26, 7352–7371. doi: http://dx.doi.org/10.1175/JCLI-D-12-00557.1

Pollard, D. and DeConto, R. M., 2012: Description of a hybrid ice sheet-shelf model, and application to Antarctica, Geosci. Model Dev., 5, 1273-1295, doi:10.5194/gmd-5-1273-2012.

Shreve, R.L. 1972. Movement of water in glaciers.J. Glaciol., 11 (62), 205–214.

---

## Author Comment (AC1) · 3 May 2016

**The role of basal hydrology in the surging of the Laurentide Ice Sheet: response to reviewers**

We thank the reviewers for their comments. We begin our response with a general defence of use of the Shallow Ice Approximation model. We then address the specific issues of each reviewer. We reproduce *the reviewers comments in blue italics*, with our comments in black.

**General Comments**

Both reviewers note, as do we in our manuscript, that the use of the Shallow Ice Approximation (SIA) may affect our results because of the omission of longitudinal stresses.

The first thing to note is that because of the horizontal grid resolution that we use (chosen to allow us to use available computer resources to probe the parameter space) the effect of longitudinal stresses are greatly diminished. Following Paterson (1983) we may assume that basal shear stress can be approximated by the gravitational driving stresses alone when averaging over distances greater than around 20 times the ice thickness. With our chosen resolution, 50 km, we are well within this limit over much of the ice sheet.

In practice the effect of the longitudinal stresses is to smooth the stress field, and thus velocities, in comparison to the SIA. In order to mimic this effect on the ice sheet we undertook a large number of tests to approximate this effect by applying a smoother to the temperature field in the model (not shown in the original m/s). This smoother followed Beuler et al. (2007) and applied a Gaussian smoothing kernel to the temperature field calculated by the model. We found that this smoother had no effect on the surging behaviour in the model: regardless of how much smoothing was applied the model surged (see attached Figure 1).

Finally, while we must accept that when the ice is surging the SIA does not encapsulate the physics of this behaviour, it has been shown (Hindmarsh 2006, Kyrke-Smith et al 2013) that more complete stress balance models do simulate the onset of surging in a manner consistent with our model. Thus, although while surging our chosen model is insufficient, it can simulate the transition into the surges.

**Comments to Reviewer Lev Tarasov.**

*Glimmer has higher order physics options. Could this not be turned on for a short (eg15 kyr) test?*

We fear that 15kyr would be insufficient time for the model to equilibrate to the different stress balance. Furthermore, it would not allow us to simulate more than one event; we would not feel comfortable reporting findings from such a limited number of events.

*The other issue is the choice of water depth dependence as opposed to basal water pressure.*

We agree that there is a discrepancy between the mm scale water depths we simulate and the metre scale roughness at the bed of ice sheets. However, we note that at the grid resolution that we are simulating such features are far below the resolution that can be resolved. To address the motivation for using water depth rather than pressure we will include the following text in Section 2.1 where we discuss the sliding scheme. "The use of water depth as the control upon fast sliding has been suggested to be a better representation than water pressure because it is the water content of the till that determines the sliding (Le Brocq et al. 2009). This parametersation, although reasonable, is, however, an empirical relationship. At present, fully process based hydrology models are not yet suitable for long-term continental scale integration and are thus unsuitable for our purposes."

The exact value of the water depth can not be known *a priori* and this was the motivation for thoroughly investigating the parameter space around the different water thresholds. Practically, until detailed hydrology models are incorporated into ice sheet models approximations such as the one that we use must be made. As an analogy: climate models must parameterise convection as it occurs at scales below that of the grid resolution, and it is only recently that simulations have begun to resolve convective processes. Let us hope that we do not have to wait as long for ice sheet models to reach this stage!

*The depth of water beneath ice sheets has been argued to be intimately related to the*
*speed with the overlying ice can slide (Budd and Jenssen, 1987; Le Brocq et al., 2009).*
*# Yes but the 2nd reference also raises the issue of how to reconcile mm scale wa-*
*ter depth with potentially metre scale water storage in subglacial sediment. It needs*
*to be made clear that this parametrization as of yet has no clear physical basis.*

See comments above.

*We assume here that the effective pressure is zero (see, e.g., Budd and Jenssen,*
*1987; Alley, 1996). Although we would expect the effective pressure to have an impact*
*upon the rate of sliding we neglect this effect as it is small.*
*#I see no basis for either of the above claims (depending what "close" means, presum-*
*ably small enough to be ignored) given current literature and understanding (eg Cuffey*
*and Patterson, 2010, for a broad review). ###########*

We shall rewrite this to read: "As a closure we shall assume that the effective pressure is zero (see, e.g., Budd and Jenssen, 1987; Alley, 1996)

*If temperatures are anomalously cold we would expect a reduction in the mass lost*
*from the ice sheet from surface melt but an increase in the mass lost due to calving.*
*# The later does not follow necessarily. Perennial landfast sea ice could choke up*
*the system as presently observed seasonally for tidewater glaciers in the Arctic. Cold*
*conditions could also reduce thermal forcings of calving. ###########*

*An increase in the calving could make it easier for the freshwater from the ice sheet to*
*impact the AMOC, but it will undoubtedly also increase the ice shelf's thickness making*
*it more resistant to melt and a better buttress.*
*# Do you mean thickness at the calving front? I would expect thickness at the grounding*
*line to decrease with increasing calving (with some time lag) due to less buttressing*
*from less shelf extent ###########*

We agree with both of these comments. Our aim in this short section was to try and construct an argument to explain why you might expect to see increased freshwater flux/calving from an ice sheet when the temperatures are anomalously cold, such as before Heinrich Events. This is an necessary argument if one believes that changes in the AMOC are linked with Heinrich Events. The literature, currently lacks such an argument, so in the interests of fairly proposing the externally forced Heinrich Event hypothesis, we attempted to construct such an argument. As is pointed out there are flaws and inconsistencies in our argument, which rather highlights the difficulty in making this link. Due to the vagueness of this argument we suggest that we remove the sentences beginning "If temperatures are anomalously cold" ending "changes in the ice shelf thickness have not been simulated (Hulbe,1997)." This paragraph will now read:

"Uncertainties surrounding an external trigger include the ultimate reason for the warming beneath the assumed ice shelf covering parts of the Labrador Sea. Although changes in the Atlantic Meridional Overturning Circulation (AMOC) have been implicated as the cause for the warming (e.g. Marcott et al., 2011; Menviel et al., 2014), it is not clear why the AMOC is itself reduced. If we assume that AMOC reduction goes hand in hand with Dansgaard/Oeschger (D/O) events, which is itself by no means certain (Dokken et al., 2013), it must be explained why the AMOC is more reduced during some D/O events than others such that a HE does not occur for each D/O event. This could arise from the link between the coldest stadials and HE. Other key features required for an external trigger also remain, so far, unobserved. Not all HEs are observed to have an associated subsurface warming, although this is due to a lack of observations rather than an evidence of absence (Marcott et al., 2011). There is also no evidence for an ice shelf in the Labrador Sea. The geography of the Labrador Sea makes it likely that an ice shelf would form there, however its size and therefore capacity to buttress the ice sheet is unknown. Observations of this ice shelf are key to supporting this mechanism."

*We acknowledge this omission but must neglect it since using higher order approximations make the long model integrations that we need to perform computationallyimpossible.*
*# Could you at least do a 40 kyr integration at 10 km, interpolating a restart file from the 25 km run to avoid the spin-up? ###########*

This experiment would be ~10 times more computationally expensive than a 25km run (2x2 times more intensive due to horizontal resolution, ~2 times more intensive due time step changes). 10 times more computer time than for the 25km runs (we did more than the one simulation presented to ensure that the results were robust) we feel is not justifiable. Furthermore we would be uncomfortable with basing our results on a single 40kyr run at 12.5km.

*At the base of the ice sheet the vertical gradient of temperature, contained in the vertical advection and diffusion terms is a result of heating by the geothermal heat flux and heating due to friction at the bed.*
*This warming is the result of the geothermal heat flux and, especially in the Hudson Strait region, the strain heating*
*# Incorrect. Basal temperature is the result of energy conservation, and is therefore due to all terms. Your figure 5 shows that "other terms" contributes more than strain heating. ###########*

Indeed basal temperature is a result of energy conservation. The temperature *gradient,* however, is due to the geothermal heat flux and heating due to friction (this is a boundary condition to the model's temperature equation).

Re the second point: the two sentences read:
"The occurrence of the events is the result of a slow warming at the base of the ice sheet that gradually brings the ice sheet bed to pressure melting point, at which time a layer of water can form at the base of the ice sheet. This warming is the result of the geothermal heat flux and, especially in the Hudson Strait region, the strain heating."

Fig 5. shows exactly, this. Before the event the two terms that are significant are the Strain Heating and the Other Terms. Once the event is under way (from 28 kyr on) the strain heating term is indeed negligible. We note that for clarity in the figures we include the geothermal heat flux in the other terms. We shall note this is the figure caption.

*Previous models have taken as the switch the temperature at the bed of the ice sheet
(Calov et al., 2002, 2010; Papa et al., 2005).
**Not all models, eg Johnson and Fastook, 2002 ###########**

We thank the reviewer for pointing this out. We shall rewrite the sentence to read:
"Previous models *looking at ice sheet surging* have taken as the switch the temperature at the bed of
the ice sheet (Calov et al., 2002, 2010; Papa et al., 2005)" and add at the end of the paragraph:
"Water depth has been shown to be important in simulating the slow evolution of ice sheets over a
glacial cycle (Johnson and Fastook 2002)"

*The behaviour of the events are broadly similar, with events being of similar size and
duration. This is strongly indicative of the robustness of the events to resolution.
**"similar size and duration" with presentation of the actual results does not provide**
any evidence of robustness. Provide a time series comparison. ###########*

See the attached Figure 2 which we shall include in the supplement.

*This compares well with the ice5g distribution that the model was initialized with, which
has an area of 1.68×107 km2.
**Not surprising, if ice5g was used as the boundary condition for the FAMOUS run**

Not surprising, but it is also not a given that this would be the case.

*At some time the base of the ice sheet will warm sufficiently that the gradient in ice
sheet surface, and its associated strain heating, can warm the interior of the ice sheet
above pressure melting point.*

*# Is the basal water flow blockage switch off/on at the pressure melting point? This
would not be physical as a 50 km square block of ice won't freeze or get warm-based
simultaneously across its base and the experiment should be repeated again with a
smooth ramped transition over some range O(0.1 to 0.5 K) ###########*

Water flow beneath the ice sheet is determined by the melt rate at the ice sheet base which in turn is
determined by the convergence of energy in the bottom model layer. There is no explicit switch that
blocks the water flow when the ice is at pressure melting point.
We shall clarify the sentence to read: "At some time the base of the ice sheet will warm sufficiently
that the gradient in ice  sheet surface, and its associated strain heating, can begin to melt the ice at
the ice sheet bed"

*As the water depth increases the sliding speed increases and thus the heating rate
from friction can increase.
**Physically, increasing water depth decreases effective basal drag to permit increased**
sliding speed, so its not clear if the heating rate from increasing water depth should
necessarily increase though it's clear why it does in the current model. ###########*

*These two regions are determined using a global sediment thickness map (Laske and
Masters, 1997).
**Caveat, this thickness map was created for a seismology context and has numerous**
errors for a glaciology context. ###########*

This is good to have documented, but unfortunately we are unaware of any other such dataset. Since
it was used merely for mapping regions with high or low sliding we do not feel it contributed any

errors. If we had used it to define our sliding parameter this would have been a far bigger concern.

*reasonably simulate sliding at the base of the present day West Antarctic Ice Sheet*
*# vague claims such as the above are common within the ice sheet modelling community, but indefensible. Be more precise. ###########*

Rewrriten: "Following Le Brocq et al. (2009) we model the onset of sliding as a tanh function of water depth which has been used to simulate sliding at the base of the present day West Antarctic Ice Sheet"

*# figure 10, need to label plots (a=, b=) so that the reader can decipher without opening up another page*

We shall add these to the caption

**Anon Reviewer:**

*SIA model comments*

For justification of use of the SIA see our opening comments. We will emphasise, that although we would have liked to use models with higher order physics we chose to probe the highly uncertain parameter space relating to the sliding parameterisation and the hydrology schemes rather than investigate higher order physics, whose effect is negligible at the model resolution we chose.

*Water scheme model comments.*
We reproduce the comments made above about the water scheme

We agree that there is a slight discrepancy between the mm scale water depths we simulate and the metre scale roughness at the bed of ice sheets. However, we note that at the grid resolution that we are simulating such features are far below the resolution that can be resolved. To address the motivation for using water depth rather than pressure we shall include the following text in Section 2.1 where we discuss the sliding scheme. "The use of water depth as the control upon fast sliding has been suggested to be a better representation than water pressure because it is the water content of the till that determines the sliding (Le Brocq et al. 2009). This parametersation, although reasonable, is however an empirical relationship. At present, fully process based hydrology models are not yet suitable for long-term continental scale integration and are thus unsuitable for our purposes."
The exact value of the water depth can not be known *a priori* and this was the motivation for thoroughly investigating the parameter space around the different water thresholds.
Practically, until detailed hydrology models are incorporated into ice sheet models approximations such as the one that we use must be made. As an analogy: climate models must parameterise convection as it occurs at scales below that of the grid resolution, and it is only recently that simulations have begun to resolve convective processes. Let us hope that we do not have to wait as long for ice sheet models to reach this stage!

*Isostasy model*

We attach a Fig 3 which shows time series similar to those in the m/s Fig 3 in which we use an elastic lithosphere model based on Lambeck and Nakiboglu (1980) and compare it to a simulation without the lithosphere model. As you can see the results are qualitatively indistinguishable from one another. The decision not to use an isostasy model was made to ensure that any changes in the events that we saw were the result of the parameter being investigated not from changes in the

bedrock topography which we would be unable to control.

*Page 7, equation (5):*

The equation should read (rho_w – rho_i ), we thank the reviewer for spotting this.

*Page 8, line 21: time interval of output*
The output is every 10 years (not every time step). We shall correct the m/s to reflect this.

*Page 8, line 24: field rescaling*
The surface mass balance fields are interpolated onto the ice sheet model grid. No lapse rate correction is applied to the temperature filed to ensure that the SMB/surface temperature field is the same for all model simulations. Thus the reported changes in the simulated surge events arise solely from the parameter being varied, not from changes in the SMB/surface temperature that might result from different ice sheet height between model runs.

*Page 8, line 27: geothermal heat flux choice*
We argue that any spatial/temporal variability in the geothermal heat flux is a small effect that will not change the overall nature of the surges, especially when compared to the effect of using a fundamentally different hydrology scheme. This is not to say that geothermal heat flux is not important, rather that its temporal/spatial variation is negligible.

*Page 8, line 31: sliding parameters*
The sliding parameter is highly uncertain, hence the numerous sensitivity tests we undertook. We will add a comment here pointing the reader to section 2.1 where  we describe why we chose these values.

*Page 8 calving parameterisation*
We use the fixed horizontal boundary condition. We shall add a comment to this effect in Section 2.5 *Further model details*

*Page 15 convergence:*
We are happy to drop this part of the sentence

*Typos:*
Will be corrected and we shall converge on a single definition of 3-D for clarity.

**Refs:**

Bueler E, et al. (2007)  Exact solutions to the thermomechanically coupled shallow-ice approximation: effective tools for verification. J Glaciology (53), 182

Hindmarsh RCA (2006)  Stress gradient damping of thermoviscous ice flow instabilities. JGR (111) B1

Kyrke-Smith T.M. et al.  (2013): Stress balances of ice streams in a vertically integrated, higher-order formulation. J Glaciology (59), 215

Le Brocq A.M. et al. (2009) A subglacial water-flow model for West Antarctica. J Glaciology. (55), 193

Paterson W.S.B. (1983): The Physics of Glaciers

[Figure]

Figure 1. Time series of ice sheet height over central Hudson Bay using the smoothing scheme with 4 different e-folding distances: (a) 240km, (b) 110km, (c) 75km, (d) 0km.

**Ice sheet height over Hudson Bay**

50 km resolution

30 km resolution

25 km resolution

**Calving Flux from Hudson Strait**

50 km resolution

30 km resolution

25 km resolution

Figure 2. Time series of Hudson Bay ice sheet height and calving flux for 3 different model resolutions 50km, 30km ,25km. Fig 3 from the text reproduced for comparison.

(a)

[Figure]

(b)

[Figure]

Figure 3. Comparison of model simulations with (a) and without (b) the isostasy model. Upper blue line ice sheet height over Hudson Bay. Lower red line calving flux from Hudson Strait (scales as per fig 3 in text).

---

## Referee Report (RR1)

**2nd Review of: *The role of basal hydrology in the surging of the Laurentide Ice Sheet**

by William H. G. Roberts, Antony J. Payne, and Paul J. Valdes

**General Impression**

Since the last version, the authors have taken a few changes to the manuscript that put their method to produce the binge-purge mechanism into a more critical light. I of course would have loved to see – even just for one parameter setup, to overcome the argument of lacking computational resources – a comparison to a model that would account for longitudinal stress transfer. That would have pointed towards a trigger of the surges by physics and leave less room for speculation about numerical artefacts in in their oversimplified ice flow as well as basal hydrological model, linked by the sliding law. Nevertheless, I think by adding these additional explanations readers even outside the relatively small community of ice sheet modelers now get the idea of the weak points within the model setup, hereby avoiding the impression that the case of the ice sheet (thermo-)dynamics leading to Heinrich events would be solved, but this paper rather delivers a new perspective to this problem. As you do not claim to investigate the surge mechanics itself, or derive quantitative values (although you actually provide numbers on calving fluxes) of fresh water fluxes, which would need a physically correct description of the outlet, I am able overcome my uneasiness to publish the paper without adding higher order model runs.

**Major points of critics**

Despite my request during the last review, you fail to give the details on the **time-step size** of your model. You updated the information only by the output-interval of 10 years. But knowing the actual time-step size is not irrelevant in order to judge about potential numerical instabilities, since – I presume you are running an explicit FD scheme – it determines your CFL condition. In particular, the statement that you test your model on different horizontal mesh sizes, but – at least that is what I extract from the test – not with different time-stepping sizes, leaves room for speculation that actually all your runs could fall into an instable regime defined by the applied time-step size and spatial resolution. Admittedly, your coarse mesh size plays into your favour for being able to apply longer time steps, but with SIA (and this is also in particular mentioned in the Bueler et al. paper from 2007) you have to clearly have an eye on the vertical direction (which I guess remains fixed in resolution). If you could give that information and additionally **drop a few lines on the CFL condition of the heat transfer equation** that applies in your model, it would improve your chain of arguments.

Concerning your newly added statement: *We argued, however, that at the grid resolution that we use this effect is likely negligible*. I do not like the word *neglibile*, as by this it easy to be misinterpreted, as it exactly is the large grid spacing in your model that contradicts the proper resolution of the physics at fast outlets and grounding lines. It should become clear that this is a consequence of your chosen resolution and not a justification for applying it. You cold rephrase: *We argued, however, that at the grid resolution that we use this effect is **not accounted for***.

**You should give more details about the newly applied smoothing**. As I understand from the Bueler et al. (2007) reference, they applied such a kernel to friction heat production (which is proportional to the shear rate, hence kind of introducing something similar to horizontal stress transfer) rather than the temperature field. Did you apply the same strategy? Else, if you really smoothed the temperatures, explain how you

applied the kernel and elaborate how you see that such a smoothing mimics the effect of longitudinal stresses (I would say it just enhances heat conductivity).

**Minor issues**

Page 5, line 24: parameteristions -> parametrization

Page 5, line 29: period is missing at the end of the line

page 11, line 34: period is missing at the end of the line

---

## Author Response (AR2)

**The role of basal hydrology in the surging of the Laurentide Ice Sheet: reivew 2 response.**

We include the comments in *italics* and our response in blue

**Comment 1:**

*Despite my request during the last review, you fail to give the details on the time-step size of your model. You updated the information only by the output-interval of 10 years. But knowing the actual time-step size is not irrelevant in order to judge about potential numerical instabilities, since – I presume you are running an explicit FD scheme – it determines your CFL condition. In particular, the statement that you test your model on different horizontal mesh sizes, but – at least that is what I extract from the test – not with different time-stepping sizes, leaves room for speculation that actually all your runs could fall into an instable regime defined by the applied time-step size and spatial resolution. Admittedly, your coarse mesh size plays into your favour for being able to apply longer time steps, but with SIA (and this is also in particular mentioned in the Bueler et al. paper from 2007) you have to clearly have an eye on the vertical direction (which I guess remains fixed in resolution). If you could give that information and additionally drop a few lines on the CFL condition of the heat transfer equation that applies in your model, it would improve your chain of arguments.*

We include the following in Section 2.3:

"We use a timestep of 1 year for all simulations, except for the simulations in which we increase the horizontal resolution, in which case we decrease the timestep to 0.5 years. This time step ensures that we satisfy the CFL condition for all simulations, including the condition for vertical velocities. Instability in the vertical can affect the evolution of temperature by vertical advection."

**Comment 2:**

*Concerning your newly added statement: We argued, however, that at the grid resolution that we use this effect is likely negligible. I do not like the word neglibile, as by this it easy to be misinterpreted, as it exactly is the large grid spacing in your model that contradicts the proper resolution of the physics at fast outlets and grounding lines. It should become clear that this is a consequence of your chosen resolution and not a justification for applying it. You cold rephrase: We argued, however, that at the grid resolution that we use this effect is not accounted for.*

We have replaced the sentence with:

"We argued that at the grid resolution that we employ, the effect of longitudinal stresses would not be felt."

**Comment 3:**

*You should give more details about the newly applied smoothing. As I understand from the Bueler et al. (2007) reference, they applied such a kernel to friction heat production (which is proportional to the shear rate, hence kind of introducing something similar to horizontal stress transfer) rather than the temperature field. Did you apply the same strategy? Else, if you really smoothed the temperatures, explain how you applied the kernel and elaborate how you see that such a smoothing*

*mimics the effect of longitudinal stresses (I would say it just enhances heat conductivity).*

We did indeed add the smoothing to the strain heating term not the temperature field, as was incorrectly stated in the m/s. We correct this and add some details to the m/s to read (page 16, line 1) :

[revised manuscript text omitted]